



# CLASSIC v1.0: the open-source community successor to the Canadian Land Surface Scheme (CLASS) and the Canadian Terrestrial Ecosystem Model (CTEM) - Part 1: Model framework and site-level performance

Joe R. Melton[1], Vivek K. Arora[2], Eduard Wisernig-Cojoc[1], Christian Seiler[1], Matthew Fortier[1], Ed Chan[3], and Lina Teckentrup[4]

[1]Climate Research Division, Environment and Climate Change Canada, Victoria, B.C., Canada
[2]Canadian Centre for Climate Modelling and Analysis, Environment and Climate Change Canada, Victoria, B.C., Canada
[3]Climate Research Division, Environment and Climate Change Canada, Toronto, Ont., Canada
[4]Max Planck Institute for Meteorology, 20146 Hamburg, Germany

**Correspondence:** Joe Melton (joe.melton@canada.ca)

**Abstract.**

Recent reports by the Global Carbon Project highlight large uncertainties around land surface processes such as land use change, strength of $CO_2$ fertilization, nutrient limitation and supply, and response to variability in climate. Process-based land surface models are well-suited to address these complex and emerging global change problems, but will require extensive de-
5 velopment and evaluation. The coupled Canadian Land Surface Scheme and Canadian Terrestrial Ecosystem Model (CLASS-CTEM) framework has been under continuous development by Environment and Climate Change Canada since 1987. As the open-source model of code development has revolutionized the software industry, scientific software is experiencing a similar evolution. Given the scale of the challenge facing land surface modellers, and the benefits of open-source, or community model, development, we have transitioned CLASS-CTEM from an internally developed model to an open-source community
model, which we call the Canadian Land Surface Scheme including Biogeochemical Cycles (CLASSIC) v. 1.0. CLASSIC contains many technical features specifically designed to encourage community use including software containerization for serial and parallel simulations, extensive benchmarking software and data (Automated Model Benchmarking ; AMBER), self-documenting code, community standard formats for model inputs and outputs, amongst others. Here we evaluate and benchmark CLASSIC against 31 FLUXNET sites where the model has been tailored to the site-level conditions and driven
with observed meteorology. Future versions of CLASSIC will be developed using AMBER and these initial benchmark results to evaluate model performance over time. CLASSIC remains under active development and the code, site-level benchmarking data, software container and AMBER are freely available for community use.





## 1  Introduction

Open collaboration has revolutionized software development leading to a proliferation of open source software (OSS) projects. Notable successes include the internet browser Mozilla Firefox, office suite LibreOffice, the GNU/Linux operating system and its derivative operating system for mobile devices, Android. OSS also has a large impact in scientific computing. A Google Scholar search for 'science open source software' reveals 4.7 million hits (accessed June 21th, 2019) indicating a high level of activity. In the field of land surface modelling there are several well-known large-scale community (OSS) models including the Joint UK Land Environment Simulator (JULES; Best et al., 2011; Clark et al., 2011), the Community Atmosphere Biosphere Land Exchange model (CABLE; Haverd et al., 2018), the Community Land Model (CLM; Lawrence et al., 2019), and the Noah Multi-Parameterization land surface model (Noah-MP; Niu et al., 2011). As recently stated by WIRED Magazine, 'Open source isn't counterculture anymore. It's the establishment' (Finley et al., 2019).

OSS development in land surface modelling presents several benefits to its participants including: 1) affordability, creating a new land surface scheme is a massive undertaking; 2) transparency, as the code is open to full scrutiny; 3) flexibility, the models are designed to be both used in their present configuration and also extended to answer new science questions; and 4) perpetuity, many users across diverse institutions help protect the code against loss, deletion, or obsolescence. However, it is not clear that open code will necessarily lead to more open science. Easterbrook (2014) outlines several barriers to sharing of code in his commentary piece on the utility of open code for more open science. While he argues that open code should lead to better quality code, as he sees it, the main barriers to code sharing include portability (ability to run the code on different platforms), configurability (model setup to perform a simulation), entrenchment (historical reasons behind code development decisions), model-data blur (processing of model inputs impact upon model outputs), and provenance (reproduction of a model result).

Here we present the Canadian Land Surface Scheme including Biogeochemical Cycles (CLASSIC; v 1.0), the successor to the coupled model framework of the Canadian Land Surface Scheme (CLASS; Verseghy, 2017) and the Canadian Terrestrial Ecosystem Model (CTEM; Melton and Arora, 2016). In developing the model framework for CLASSIC, careful attention has been paid to exploit the benefits of OSS, while minimizing the risks outlined by Easterbrook (2014). The CLASSIC model framework includes several key features designed to encourage collaboration and community use including: 1) self-documenting code, 2) version control allowing source code management, distributed non-linear workflows, issue tracking, and wiki functionality, 3) native support for Network Common Data Format (netCDF) input and output along with conversion tools for ASCII legacy inputs, 4) ability for code to run both serially on personal computers and using message passing interface (MPI) on computing clusters, 5) output file description and metadata handled via a web interface, 6) model parameters read in





from an external file, 7) containerization, and 8) extensive benchmarking. Each of these features will be expanded upon in the following sections.

In Section 2 we describe the submodels, CLASS and CTEM that form the scientific basis of CLASSIC and the initial state of the model framework at the start of CLASSIC development. Section 3 details the model developments implemented along with

tools to help existing users migrate to the new model framework. Section 4 outlines our model evaluation and benchmarking approach while Section 5 presents the present state of CLASSIC as evaluated by the AMBER protocol. Section 6 describes future technical and scientific directions for CLASSIC.

## 2 Model Description

### 2.1 Model physics: CLASS

The Canadian Land Surface Scheme (CLASS) was initiated in 1987 to produce a 'second generation' land surface scheme for inclusion in the Canadian general circulation model (GCM) and has been under continual development since. The first publications introducing CLASS described the physics calculations for movement of heat and water through the soil and snow layers (Verseghy, 1991), and the physics algorithms for energy and moisture fluxes within the vegetation canopy as radiation and precipitation cascade through it, with an explicit thermal separation of the vegetation from the underlying ground

(Verseghy et al., 1993). Development of CLASS has been predominantly within Environment Canada, a federal department of the Government of Canada (later renamed Environment and Climate Change Canada; ECCC) with the exception of an organized community effort as part of the Canadian Climate Research Network (1994 - 1997;  Verseghy, 2000) as well as ad-hoc collaborations with the broader research community. CLASS is presently at version 3.6.2 (Verseghy, 2017). As a land surface scheme, CLASS simulates the fluxes of energy, momentum, and water between the atmosphere and land surface (Figure

1).

CLASS operates on a sub-daily time-step that varies depending on its application. When within the framework of the most recent version of the Canadian Earth System Model (CanESM5;  Swart et al., 2019), it operates at a time step of 15 minutes, consistent with the atmospheric component. A time-step of 30 minutes is used for offline, uncoupled simulations where it is driven with observed meteorological data. Vegetation characteristics are described by rooting depth, canopy mass, leaf area

index (LAI), and vegetation height, which are then used in calculations of the transfers of energy, water, and momentum with the atmosphere. The physical land surface processes in CLASS are modelled using plant functional types (PFTs). Presently, global vegetation in CLASS is represented using four PFTs - needleleaf trees, broadleaf trees, crops and grasses. The structural vegetation characteristics of these PFTs are modelled as a function of driving meteorological data and atmospheric $CO_2$ concentration by CTEM, which handles the biogeochemical processes, as explained in the next section. The number of ground

layers in CLASS can vary depending upon application. The standard offline model setup currently uses twenty ground layers starting with 10 layers of 0.1 m thickness, gradually increasing to a 30 m thick layer for a total ground depth of over 61 m while three ground layers with thicknesses of 0.1, 0.25 and 3.75 m are used in CanESM5. Water fluxes are calculated for soil layers within the permeable soil depth of the ground column, but not the underlying bedrock layers, whereas temperatures are



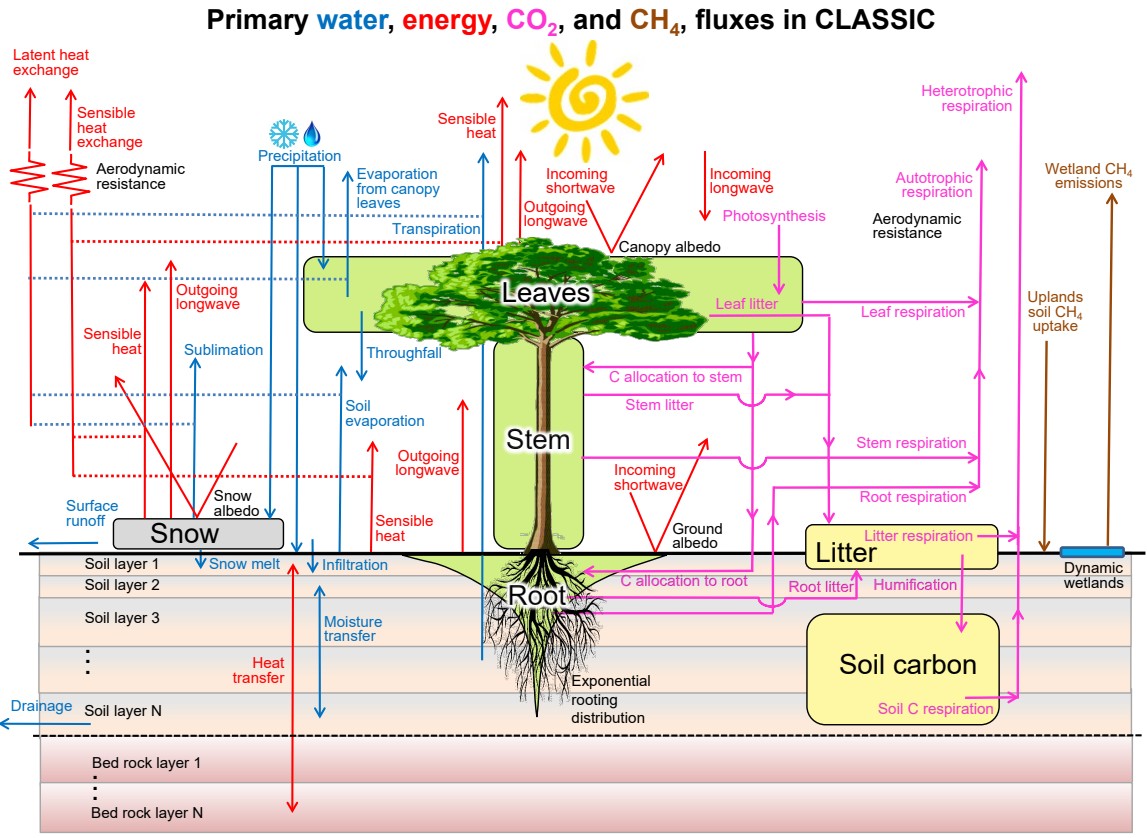

**Figure 1.** Schematic illustration of processes simulated by CLASSIC.

calculated for both soil and bedrock layers (depicted in Fig. 1). Both water fluxes and ground temperatures are calculated each time step. The permeable soil depth varies geographically and is read in from the model initialization file. Also calculated each time-step are the temperature, mass, albedo, and density of the single layer snow pack (where it exists), the temperature and interception and storage of rain and snow on the vegetation canopy, and the temperature and depth of ponded water on the soil

5   surface. Each grid cell is independent with no lateral transfers of heat or moisture between them. While runoff can be routed to estimate streamflow, such as in CanESM5 using the approach of Arora et al. (1999) or the MEC-Surface & Hydrology System (MESH) framework (Pietroniro et al., 2007), in both cases, once the runoff leaves a grid cell, it does not modify the soil moisture of downstream grid cells.

Application and evaluation of the performance of CLASS over the course of its development has been extensive with dozens

10  of publications using the model at point, regional, and global scales both in coupled and offline modes. CLASS has been





applied in an offline context, i.e. forced with observed meteorology (e.g., Bailey et al., 2000; Kothavala et al., 2005; Roy et al., 2013; Bartlett et al., 2006; Brown et al., 2006; Wu et al., 2016; Verseghy and MacKay, 2017; Melton et al., 2019d), as the physical land surface component of regional climate models, e.g. CRCM (Ganji et al., 2015; Paquin and Sushama, 2014) and CanRCM (Scinocca et al., 2016), and integrated into each version of the Canadian Atmospheric Model (CanAM;  von Salzen

et al., 2013), Coupled Global Climate Model (CanCM), and Earth System Model (CanESM;  Arora et al., 2011; Swart et al., 2019) since the early 1990s.

## 2.2 Model biogeochemistry: CTEM

Development of the Canadian Terrestrial Ecosystem Model (CTEM) began in the early 2000s at Environment Canada in response to the need for a land surface carbon cycle component for the CanESM. CTEM, which is presently at version 2.0

(Melton and Arora, 2016), couples with CLASS through the exchange of information describing the state of the physical land surface and overlying vegetation. CLASS provides CTEM with physical land surface information including soil moisture, soil temperature and net radiation. CTEM uses this information to simulate photosynthesis and prognostically calculate carbon in its three live vegetation components (leaves, stem, and roots) and two dead carbon pools (litter and soil). The carbon amounts in these five carbon pools evolve prognostically, in the default CLASSIC configuration, for nine PFTs that map directly onto

the four PFTs used by CLASS. Needleleaf trees are divided into their deciduous and evergreen types, broadleaf trees are divided into cold deciduous, drought deciduous, and evergreen types, and crops and grasses are divided based on their photosynthetic pathways into $C_3$ and $C_4$ versions. The finer distinctions between PFTs in the CTEM PFTs is required for modelling biogeochemical processes. For instance, simulating leaf phenology prognostically requires the distinction between deciduous and evergreen versions of broadleaf trees. However, once the LAI has been dynamically determined by CTEM, CLASS only

needs to know that this PFT is a broadleaf tree since the physics calculations do not require information about underlying the deciduous or evergreen nature of the leaves. The prognostic carbon masses of leaf, stem, and root simulated by CTEM are used to calculate the structural vegetation characteristics required by CLASS: rooting depth (using a dynamic root distribution; Arora and Boer, 2003), canopy mass, LAI, and vegetation height (Arora and Boer, 2005a). Other than these structural vegetation attributes, CTEM also provides canopy conductance values to CLASS based on photosynthesis calculations at the CLASS

time step, as explained in Arora and Boer (2003). The remainder of the biogeochemical process simulated by CTEM (Figure 1) and the resulting vegetation dynamics operate on a daily time step. CTEM models all primary terrestrial ecosystem processes including maintenance and growth respiration (Arora and Boer, 2005a); heterotrophic respiration (Melton et al., 2015); tissue turnover, allocation of carbon, and phenology (Arora and Boer, 2005a); disturbance (fire;  Arora and Boer, 2005b; Arora and Melton, 2018); competition for space between PFTs (Arora and Boer, 2006; Melton and Arora, 2016), and land use change

(Arora and Boer, 2010).

CTEM also dynamically calculates wetland extent, methane emissions from wetlands and fires, and methane uptake by soils (Curry, 2007) as described in Arora et al. (2018). To determine the wetland extent, as the liquid soil moisture in the top soil layer increases above latitudinally dependent threshold values, the wetland fraction increases linearly up to a maximum value, equal to the flat fraction in a grid cell with slopes less than 0.2%. Methane emissions from wetlands are calculated by scaling the





heterotrophic respiration flux from the model's litter and soil carbon pools. A separate submodule simulates peatland specific processes following Wu et al. (2016).

Both CLASS and CTEM have the capability to be run in a 'mosaic' configuration in which grid cells are divided into separate and independent tiles that simulate their own energy and water balance. There is no lateral transfer of energy or water

between tiles that lie within a grid cell. As a result, the soil moisture in the permeable soil layers and the temperature of soil and bedrock layers in each tile evolve independently of each other, despite being driven with same meteorological data. A grid cell may be divided into tiles using any criteria, e.g. by each PFT on its own tile, different soil textures, or peatlands vs. uplands. In contrast, with the 'composite' or single-tile approach, structural vegetation attributes (LAI, rooting depth, vegetation height, canopy mass, and albedo) are averaged over all PFTs for use in energy and water balance calculations. Soil texture for

permeable soil layers is also common to all PFTs within a grid cell resulting in a single liquid and frozen soil moisture for each permeable soil layer, and a single temperature for each soil and bedrock layer for the grid cell. The effect of tiling on the basis of fractional coverage of PFTs has been evaluated on estimation of the terrestrial carbon sink (Melton and Arora, 2014), competition between PFTs (Shrestha et al., 2016), and the use of soil texture clusters as a tiling criteria has been evaluated by Melton et al. (2017).

Terrestrial biogeochemical processes simulated by CTEM have been evaluated at scales from site-level to global (e.g. Peng et al., 2014; Melton and Arora, 2014; Melton et al., 2015; Melton and Arora, 2016). The fire subroutine has been extensively evaluated as part of the Fire Model Intercomparison Project (FireMIP; Hantson et al., 2016; Forkel et al., 2019) in addition to being used to estimate carbon cycle implications of the reduction in global wildfire since the 1930s (Arora and Melton, 2018). The parameterization for competition between PFTs has been evaluated at the site-level (Shrestha et al., 2016) as well as at the

global scale (Melton and Arora, 2016). CLASS-CTEM has contributed to the Global Carbon Project's methane assessment by providing wetland methane emissions (Poulter et al., 2017). An assessment of natural methane emissions simulated by CTEM is presented in Arora et al. (2018) who use a one-box model of atmospheric methane together with prescribed anthropogenic and geological sources to reproduce atmospheric methane concentrations consistent with the observational record over the historical period. CLASS-CTEM has also contributed regularly to the Global Carbon Project's annual carbon budget analyses

since 2016 (Le Quéré et al., 2016, 2018b, a).

## 3 Model development

### 3.1 Motivation

At the start of our project to transform CLASS-CTEM into CLASSIC, the model code base was not well-suited to modern computing and model development practices. CLASS-CTEM was written following the Fortran 77 standard, documentation was

provided in stand-alone documents that often significantly lagged model development or scientific papers, code management did not use modern version control systems, model parameters such as the number of soil layers and PFTs were hard-coded into subroutines, and the offline framework (as opposed to the version coupled into the family of CanESM models) used fixed format ASCII text files for model inputs and outputs. While these issues are not unique to CLASS-CTEM, they made model de-





velopment and evaluation challenging. For example, while ASCII inputs are reasonable for site-level simulations, they quickly become difficult to prepare and manage over large modelling domains, which entailed handling thousands of ASCII files that are poorly suited to parallel computation. Additionally as computer hardware or compiler versions change the model results are susceptible to issues of reproducibility due to a changing computational environment. While CLASS-CTEM has been used

extensively by the Canadian research community, as evidenced by the publications cited in the previous section, the modelling framework was poorly designed for new users who faced a steep learning curve in attempting to run and understand the model. In our modernization and improvement of the CLASS-CTEM code base to develop the CLASSIC model, we have addressed these concerns and made several important changes to the model code, which we describe below.

## 3.2   Model developments to encourage community use

### 3.2.1   Containerization

A virtual machine (VM) emulates a physical computer allowing different operating systems to run on different hardware, for e.g., the Linux operating system on a Windows machine. A software container is a form of operating system virtualization, essentially a pared-down version of a virtual machine with only the necessary software required to complete the intended task. Both options have been used to facilitate the running of complex models. The Weather Research and Forecasting model (WRF;

Hacker et al., 2016) has been implemented in both VM and container configurations, while the United Kingdom Chemistry and Aerosols (UKCA) composition-climate model has been implemented within the Met Office VM (Abraham et al., 2018). In choosing between a VM or container to provide users of CLASSIC, we chose a container due to its light computational overhead, and design geared to high performance computing (HPC) applications (Arango et al., 2017). The advantages of containers for scientific computing are numerous. Firstly, as the environment within the container is consistent across computer

systems, the reproducibility of the code is enhanced. Containers are encapsulated portable environments that can contain both the software and data dependencies, along with libraries needed by the software, ensuring all necessary environment variables are provided and consistent. Secondly, they prevent orphaning of old code due to environment changes. If a container image is retained, the container can recreate the same environment for the code each time it is run. This ability allows a simple means to rerun old code, without having to consider environment changes that would make running the code outside of the container

onerous. Third, the use of a container makes it easier to get up and running with a model quicker as all dependencies are included in the container. Lastly, many HPC centres allow the use of containers on their systems reducing the need for users to reconfigure models for different server systems.

    For CLASSIC, we use a Singularity container (https://www.sylabs.io/), which is portable to Linux, Mac, and Windows systems as well as HPC clusters, for which Singularity was specifically designed (Kurtzer et al., 2017). Tests of Singularity's

performance on HPC benchmarks have demonstrated a negligible performance overhead (Le and Paz, 2017). Singularity is already installed on many national, university and government HPC centres including Compute Canada, Argonne National Labs, Fermilab, National Supercomputer Centre (Sweden), amongst others. The CLASSIC container image is available from our Zenodo community page (https://zenodo.org/communities/classic/). Version 1.0 of our container contains all software





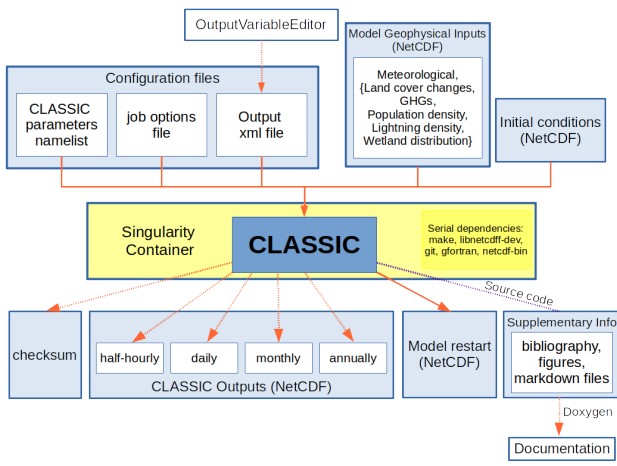

**Figure 2.** CLASSIC model workflow. For each CLASSIC simulation the solid lines indicate mandatory model inputs or outputs. The OutputVariablesEditor can be used to edit the XML file that controls the model outputs and specifies their metadata (Section 3.2.8). Model parameters are read in from the CLASSIC parameters namelist file while the simulation options such as location of input files and model configuration options are read in from the CLASSIC job options file. The required model geophysical inputs vary depending on the model configuration chosen but meteorological inputs are always required. All model outputs can be switched on or off depending on the simulation except the model restart file. Checksums can be used to ensure changes to the model don't impact model performance (see Section 3.2.2). The CLASSIC source code is combined with other supplementary information by Doxygen to produce the model documentation (see Section 3.2.3).

libraries needed to run the model in serial or parallel (using MPI) as well as the capability to run our benchmarking software (described in Section 4.2). The general model workflow is outlined in Figure 2.

### 3.2.2 Code design and management

To permit flexibility in the code, CLASSIC's parameters are read in at run time from a Fortran namelist file. This use of a
5   namelist permits rapid testing of model sensitivity to parameter values, as the model does not require recompilation between tests. Additionally, new PFTs can be more easily added to the code as all parameters are located in the namelist file, rather than distributed throughout the code. However, by design, new PFTs cannot be introduced into CLASSIC without due care. Within the code, case structures are used at each PFT-level branching calculation to determine the code branch a particular PFT is assigned to. These case structures have integrated error checks, whereby an unknown PFT causes the model to abort with a
10   flag thrown informing the user where in the code the model detected an unknown PFT. This safety check helps to encourage thoughtful introduction of new PFTs to the model.

In transitioning from CLASS-CTEM to CLASSIC, the code base was ported into the distributed version control system Git (https://git-scm.com/) and is distributed via the software development tool Gitlab (https://gitlab.com/cccma/classic). The Gitlab





issues tracker and a Nabble forum (http://classic-message-board.158658.n8.nabble.com/) will be used to facilitate communication between CLASSIC users. The issues tracker can be used to submit bug reports, elaborate on new parameterizations, and discuss code changes whereas the forum can be used to ask other users for advice and assistance.

As a further aid to development, CLASSIC can output checksums to speed model development for code changes that should not impact upon the model outputs. The checksum subroutine computes content-based checksums (summation of the flipped bits, i.e. the 1 in binary representation) of several groups of variables after a run has completed.

Checksums are an imperfect means of ensuring no logical changes, as two numbers may have different binary representations with the same number of flipped bits. However, as the number of variables checked increases, it becomes highly unlikely to render a false-positive. For example, if we take a data item of bit-length $n$, having $b$ flipped bits in its representation, the number of same-length data items with the same checksum can be expressed through the binomial coefficient

$$\binom{n}{b}.$$
(1)

If we assume the worst case where $b = \frac{n}{2}$, then Equation 1 becomes

$$\binom{n}{n/2}.$$
(2)

Dividing by the total number of possible values for an $n$-digit binary number, we find the probability of a false positive checksum to be:

$$\frac{\binom{n}{n/2}}{2^n}.$$
(3)

So, for example, if the checksum is computed using only three 32-bit numbers, the probability of a false positive is about $2.87 \times 10^{-7}$.

### 3.2.3 Self-documenting Code

To modernize the CLASS-CTEM code base, it was first transformed from the fixed form (Fortran 77) to a free-form structure (allowing coding constructs permitted by the Fortran 2008 standard). Best practices suggest that it is most desirable to embed documentation within the software, as this increases the likelihood that when developers update code, they will update the documentation at the same time (Wilson et al., 2014). Our model documentation was then incorporated directly into the code using the syntax of the Doxygen documentation generator (http://www.doxygen.nl/). The use of Doxygen allows both the programmatic and scientific reasoning behind the code to be detailed within the generated manual along with variable dictionaries. This ensures that the documentation for a particular model version is always included in the model source code distribution, and lowers the burden for developers to maintain its currency as it can be edited as the source code is updated. The full Doxygen-generated CLASSIC documentation for version 1.0 can be found in the Zenodo archive and, for the most





recent CLASSIC release, at https://cccma.gitlab.io/classic. As Doxygen is included in the CLASSIC container, the user can run 'doxygen Doxyfile' to generate a local version of the documentation as required.

### 3.2.4 Coding standards

As part of the code modernization effort, we refactored the code to follow a recently developed CLASSIC coding convention. Coding conventions help to ensure code portability, readability, and maintainability. In adopting our coding conventions, we have developed a tool (a 'linter') to enforce code quality that can be used on legacy code, or new code, to either change the code to meet the coding standards, or to flag suspect sections of the code for manual intervention. Our coding convention is available in the Supplementary Material and on the CLASSIC website. The linter, which is written in python, is available in the CLASSIC code repository.

### 3.2.5 Serial and parallel computations (MPI)

As CLASSIC can be run at the site-level as well as over gridded domains it is important to support both modes of operation while maintaining only one codebase. The newly created CLASSIC driver allows compilation for serial or parallel processing of the model code. For site-level simulations, serial compilation and running of the model is sufficient whereas, for speed, runs over gridded domains require parallel processing which uses message passing interface (MPI) directives within the CLASSIC code. Pre-compiler directives, used sparingly, allow the same code for both applications. The default compiler supplied within the CLASSIC container is the GNU compiler (gfortran for serial and mpif90 as wrapper around gfortran for parallel computation) with additional Makefile scripts for Intel and Cray compilers.

### 3.2.6 Simulation domains

CLASSIC is designed to be easily run over simulation domains from site-level to global. To run the model at a point location, the model input and output files can be either, also site-level with point-scale information, or regional (two-dimensional fields). For regional domains, the model grid can be specified either with one-dimensional vectors or two-dimensional grids of longitudes and latitudes. The 1D case includes regular grids equally spaced in longitude and latitude as well as regular Gaussian grids. The 2D case is more general and can support any rectangular grid. The model call determines whether a simulation is run across a region or at a point. When using a regular grid, the model domain can be set by calling CLASSIC with a longitude latitude box. For example, '-180.0/180.0/-90.0/90.0' specifies a global simulation, '-135.0/-105.0/45.0/90.0' a region encompassing western Canada, and '-105.0/50.5' a grid cell centred on 105 °W and 50 °30'N. When the input files are provided on an irregular grid, CLASSIC uses grid cell indexes, instead of geographic coordinates to delineate the domain for the simulation. Thus '-135.0/-105.0/45.0/90.0' could be specified on an irregular grid by the grid cell indexes of those corresponding coordinates, i.e. '39/150/50/150'.





### 3.2.7   NetCDF input/outputs

While CLASS-CTEM used ASCII text files for model input/output (I/O), this method is cumbersome over gridded domains. The CLASSIC framework uses netCDF files instead (https://www.unidata.ucar.edu/software/netcdf/. NetCDF is a machine independent data format that is self-describing, allowing extensive metadata within the files, and which is the most common

data format within the land surface modelling community. Scripts are provided in the CLASSIC code repository to convert legacy ASCII input files (meteorology and model initialization files) into the new netCDF format. Additionally, for site level users, a Fortran tool is included to convert Fortran namelist files, which are easier to work with than the legacy ASCII files, into a netCDF file suitable for CLASSIC model initialization.

### 3.2.8   Output files (xml)

To best use netCDF as an output format requires writing the output file metadata at the time of the file's creation. To facilitate this CLASSIC uses extensible markup language (XML) files that are edited through a web interface and the resulting files are validated using an adjacent schema. The web interface allows a user to configure the output variables, as well as add new variables and edit metadata, using any JavaScript-compatible browser. Once the changes are complete, the user may download an updated version of the modified XML configuration file for use by CLASSIC. Where possible CLASSIC output files are

Climate and Forecast (CF)-compliant (http://cfconventions.org/) and use variable names consistent with the data request of the Coupled Model Intercomparison Project (CMIP6; http://clipc-services.ceda.ac.uk/dreq/mipVars.html).

### 3.2.9   Meteorological inputs

Meteorological inputs required by CLASSIC include downwelling shortwave and longwave radiation, surface precipitation rate, surface air pressure, specific humidity, wind speed, and air temperature together with a reference height at which these

quantities were measured. Within the CanESM framework these inputs are provided on the coupled model physics timestep of 15 minutes. For global offline simulations, meteorological variables are typically available on either a 3 hour or 6 hour timestep. To convert the 3/6 hourly meteorological data to the offline CLASSIC physics timestep of 30 minutes, CLASSIC disaggregates the coarse temporal resolution meteorology on the fly. Surface pressure, specific humidity, wind speed and surface temperature are linearly interpolated. Long-wave radiation is uniformly distributed across the 3/6 h period. Shortwave

radiation is distributed diurnally using the day of year and a grid cell's latitude with the maximum value occurring at solar noon. The total 3/6 h precipitation amount determines the number of wet half-hours in each 3/6 h period following Arora (1997). In a conservative manner, the total 3/6 h precipitation amount is then randomly spread across the wet half-hourly periods. If CLASSIC is being run with observed meteorology, such as from an eddy covariance tower, on a 30 minute timestep, the meteorology is used without modification.





## 4  Running and Benchmarking CLASSIC

### 4.1  Quick-start tutorial to run and benchmark CLASSIC

Appendix A contains a tutorial that guides the reader through downloading, compiling, and running CLASSIC over a set of FLUXNET sites. FLUXNET is a global network of micrometeorological tower sites that uses the eddy covariance method

to measure the exchanges of carbon dioxide, water vapour, and energy between terrestrial ecosystems and the atmosphere (https://fluxnet.ornl.gov). The outputs of the model runs are then run through a benchmarking system described below. All figures presented here except the schematic figures (Figs. 1 and 2) are produced by the benchmarking software.

### 4.2  Automated Model Benchmarking (AMBER)

The increasing complexity of land surface models necessitates more advanced methods of model evaluation and assessment.

Such methods have been developed in a number of collaborative projects including the Project for Intercomparison of Land Surface Parameterization Schemes (PILPS;  Henderson-Sellers et al., 1993), the Global Land-Atmosphere Coupling Experiment (GLACE; Koster et al., 2006), the Protocol for the Analysis of Land Surface Models (PALS; Abramowitz, 2012), and PALS Land Surface Model Benchmarking Evaluation Project (PLUMBER; Best et al., 2015), and the International Land Model Benchmarking (ILAMB; Collier et al., 2018).

ILAMB has developed a framework that summarizes model performance across multiple statistical metrics using a dimensionless skill score system implemented in an open-source benchmarking and diagnostics software tool for land surface model evaluation written in Python (Collier et al., 2018). The evaluation of CLASSIC presented in this study is based on ILAMB's statistical framework, which we implemented in a new R package referred to as the Automated Model Benchmarking (AMBER) package (Seiler, 2019). The development of AMBER allowed us to tailor the ILAMB approach to CLASSIC model

outputs allowing: ($i$) a seamless data ingestion that does not require pre-processing steps, ($ii$) the ability to evaluate CLASSIC in different simulation modes (i.e. global and regional simulations on differing grids, and site-level runs), and ($iii$) full control on how the statistical framework is implemented. AMBER uses a variety of observation-based reference data against which it evaluates CLASSIC. These data consist of global scale remote sensing-based products, eddy covariance flux towers and annual streamflow measurements at the mouth of major rivers. Site-level evaluations are based on eddy covariance flux tower data

alone. All AMBER outputs from site-level evaluation of CLASSIC v. 1.0 are included in the model benchmarking archive (Table 2).

#### 4.2.1  Skill scores

In ILAMB (Collier et al., 2018) and AMBER the performance of a model is expressed through scores that range from zero to one, where higher values imply better performance. These scores are computed for each variable in five steps: (1) computation

of a statistical metric, (2) nondimensionalization, (3) conversion to unit interval, (4) spatial integration, and (5) averaging scores





computed from different statistical metrics. The statistical metrics considered are the bias, root-mean-square error, phase shift, interannual variability, and spatial distribution. For example, the scalar score for the bias is calculated as:

$$bias(\lambda, \phi) = \overline{v_{mod}}(t, \lambda, \phi) - \overline{v_{ref}}(t, \lambda, \phi), \tag{4}$$

where $\overline{v_{mod}}(t, \lambda, \phi)$ and $\overline{v_{ref}}(t, \lambda, \phi)$ are the mean values in time ($t$), over a specified period of time, of a variable $v$, which

varies geographically as a function of longitude $\lambda$ and latitude $\phi$ for model and reference data, respectively. Nondimensionalization is achieved by dividing the absolute bias by the standard deviation of the reference data ($\sigma_{ref}$) that corresponds to the same period over which the mean of model simulated quantities is calculated:

$$\varepsilon_{bias}(\lambda, \phi) = \frac{|bias(\lambda, \phi)|}{\sigma_{ref}(\lambda, \phi)}. \tag{5}$$

A bias score that ranges between zero and one is calculated next:

$$s_{bias}(\lambda, \phi) = e^{-\varepsilon_{bias}(\lambda, \phi)}. \tag{6}$$

The spatial integration of $s_{bias}$ leads to the scalar score:

$$S_{bias} = \overline{\overline{s_{bias}}}. \tag{7}$$

The spatially integrated score values for the root-mean-square error ($S_{rmse}$), phase shift ($S_{phase}$), interannual variability ($S_{iav}$), and spatial distribution ($S_{dist}$) are found in a similar way although the actual calculation of the metrics is, of course,

different as shown in Table A1. To demonstrate the intermediate statistical metrics for the component scores, the components of $S_{rmse}$ are listed in Table A2. The score values based on these metrics are then combined to derive a single overall score for each output variable:

$$S_{overall} = \frac{S_{bias} + 2S_{rmse} + S_{phase} + S_{iav} + S_{dist}}{1 + 2 + 1 + 1 + 1}. \tag{8}$$

$S_{rmse}$ is assigned twice as much value as the other metrics since we consider it more important than the other metrics.

Interpretation of benchmarking scores should be done carefully. While ILAMB uses 'stoplight' colours to distinguish different thresholds, at least in a visual sense, of model performance (e.g. see Figure 1 in Collier et al., 2018), we don't adopt that approach. The use of thresholds may be a useful visual tool when distinguishing the performance of several models (as in Figure 1 of Collier et al., 2018), or versions of the same model, however here we are presenting the results of a single model and thus any thresholds chosen would be arbitrary. Following Collier et al. (2018) we also '... do not view these aggregate absolute

scores as a determinant of *good* or *bad* models. We envision the scores as a tool to more quickly identify relative differences among models and model versions which the scientist must then interpret'. We also agree with Collier et al. (2018) that the





absolute value of score is not particularly meaningful. As well, a perfect score is not achievable due to reference datasets having measurement error and uncertainty, a lack of consistency between datasets for the same variables, and a lower score may only highlight the need for an enhanced observational sampling effort, or the lack of an appropriate metric of model performance (Collier et al., 2018). Therefore, we do not use the model score as the basis to determine whether model performance is ac-

ceptable or deficient for different model outputs, but rather use these scores to define the initial model performance that future model developments will be compared against.

## 5 Evaluation of CLASSIC

We used three separate approaches to benchmark CLASSIC. First, we used FLUXNET sites (Pastorello et al., 2017) where CLASSIC was driven with observed meteorology and the model initialization file was set up to correspond to site conditions,

i.e. vegetation composition and coverage, soil texture and permeable depth, based on publications describing the tower sites. These 'site-level' simulations were performed for the 31 FLUXNET sites listed in Table 1 and shown in Figure 3. We then evaluated the CLASSIC outputs against the eddy covariance (EC) tower derived quantities of net ecosystem exchange (NEE), gross primary productivity (GPP), ecosystem respiration (RECO), net radiation (RNS), ground (HFG), sensible (HFSS) and latent heat (HFLS) fluxes. The FLUXNET data is from the November 3rd, 2016 update to the FLUXNET2015 dataset. While

these sites are limited in their spatial and temporal coverage, they do present the most realistic model setup which uses observed vegetation composition, soil textures and driving meteorology.

 There are two limitations, which apply to most land surface models, when comparing model simulated data against observation-based estimates at FLUXNET sites. First, CLASSIC represents vegetation at the level of PFTs, and not at the species level. That is, for example, at two temperate FLUXNET sites which have either Oak (*Quercus*) or Maple (*Acer*) deciduous broadleaf

trees, CLASSIC represents these sites using the same set of model parameter values corresponding to its cold deciduous broadleaf PFT. While climate predominately determines energy, water, and carbon fluxes, in the real world, species level differences also play a role in modulating these fluxes. Second, the model simulated values are compared with observations only when the model's carbon pools reach equilibrium after being driven repeatedly with available meteorological data. The chosen FLUXNET sites in our study have between 3 and 19 years of available meteorological data (Table 1). Since these available

years of meteorological data may not be representative of the mean climate at a given site, and/or a given site may be recovering from events that occurred prior to the start of the EC tower installation, the comparison between observations and model simulated fluxes is somewhat confounded. The modelled annual NEE thus always sums to zero, by construction, while we know that in the real world, at most sites, GPP is currently higher than RECO making annual NEE positive with land acting as a sink of carbon in response to rising atmospheric $CO_2$ concentrations. Past episodic extreme climatic events (e.g. drought or

extreme precipitation) and disturbances (e.g. prior timber harvest or tree planting such as site CZ-BK1 (Sedlák et al., 2010)) or fire, e.g. at CG-Tch (Merbold et al., 2008)) that we are unable to take into account confound the mismatch between model and observation-based fluxes even more. In an ideal world, we would have historical meteorological data and disturbance history





available at each site. All components of the carbon budget - GPP, RECO, and NEE - are affected by these issues, but since NEE is a residual of GPP and RECO, the model to observations comparison is confounded most significantly for NEE.

The other two benchmarking approaches drive CLASSIC with globally gridded data products of meteorological data, vegetation cover, soil permeable depth, and soil textures. In the first of these two approaches, CLASSIC is evaluated against globally

gridded data sets for variables where observation-based data are available. The third and final benchmarking approach also uses gridded data products to drive CLASSIC, but the model outputs are evaluated against the entire FLUXNET2015 release of 212 sites by comparing the EC tower data against the model outputs from the CLASSIC grid cell that each tower lies within. The evaluation from these two approaches is presented in a companion paper that evaluates CLASSIC on a global scale (Seiler et al. 2019, in prep.).

## 5.1   Further performance metrics

We present here also the statistical metrics of root mean squared error and coefficient of determination (Both calculated using the scikit-learn python package; Pedregosa et al., 2011). The coefficient of determination is calculated as:

$$R^2(y, \hat{y}) = 1 - \frac{\sum_{i=1}^{n}(y_i - \hat{y}_i)^2}{\sum_{i=1}^{n}(y_i - \bar{y})^2} \tag{9}$$

where $\hat{y}$ is the predicted value of the $n$-th sample and $y_i$ is the corresponding observed value for total $n$ samples. $\bar{y}$ is found

via

$$\bar{y} = \frac{1}{n} \sum_{i=1}^{n} y_i \tag{10}$$

and

$$\sum_{i=1}^{n}(y_i - \hat{y}_i)^2 = \sum_{i=1}^{n} \epsilon_i^2 \tag{11}$$

While the coefficient of determination is commonly termed $R^2$, in this formulation, the best possible goodness of fit is 1.0

while the $R^2$ value can go *negative* (as can be seen from Equation 9). The RMSE is calculated by

$$\text{RMSE}(y, \hat{y}) = \sqrt{\frac{1}{n} \sum_{i=0}^{n-1}(y_i - \hat{y}_i)^2}. \tag{12}$$

## 5.2   Benchmarking observation-driven CLASSIC at FLUXNET sites

The overall AMBER scores for the comparison between the observation-driven CLASSIC outputs and the 31 FLUXNET sites are shown in Figure 4 for three energy fluxes (HFLS, HFSS, and RNS) and three carbon fluxes (GPP, NEE, and RECO). For the



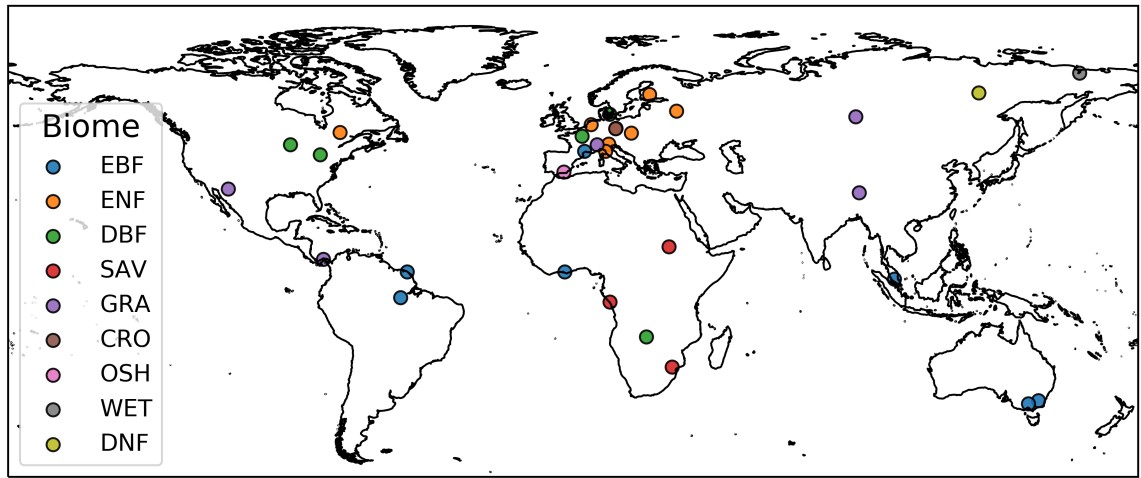

**Figure 3.** Locations of the 31 FLUXNET sites grouped by biome (Table 1). International Geosphere-Biosphere Programme (IGBP) land classification abbreviations used include Evergreen Broadleaf Forests (EBF), Evergreen Needleleaf Forests (ENF), Savannas (SAV), Deciduous Broadleaf Forests (DBF), Grasslands (GRA), Croplands (CRO), Open Shrublands (OSH), Closed Shrublands (CSH), Permanent Wetlands (WET), and Deciduous Needleleaf Forests (DNF).

component scores, CLASSIC generally had higher phase and spatial distribution scores with lower scores for RMSE, which is weighted double the other metrics in calculating the overall score (see Section 4.2.1). Most overall scores are between 0.65 - 0.82 with the exception of NEE at 0.44. These scores are discussed more below.

### 5.2.1 Energy fluxes

The mean seasonal cycle of net radiation (RNS) simulated by CLASSIC compared against observation-based estimates from all FLUXNET sites is shown in Fig. 5. For each site, the plot title shows the site ID (Table 1) followed by the biome type in brackets. The average RMSE across the sites is 19 W m$^{-2}$ with a median R$^2$ of 0.86 (see Supplementary Table S1 for per site values). Sites with a pronounced seasonality, typically from the high latitudes, are better captured by CLASSIC compared to lower latitude sites where the seasonal cycle is a result of seasonality in cloudiness and precipitation. CLASSIC also generally

has lower peak values than observations, which could indicate that the simulated albedo values are too high. The most poorly simulated sites include BR-Sa1 (EBF), ZM-Mon (DBF), RU-Che (WET), and RU-SkP (DNF). These sites, with the exception of BR-Sa1, have only 2 – 3 years of observations. The AMBER scores for RNS are all above 0.73 with an overall score of 0.82 and an RMSE score of 0.76, which are the highest scores for the energy fluxes.

The time series of the observed and simulated latent heat fluxes (HFLS) is shown in Figure 6. Only 23 of the 31 FLUXNET

sites reported latent and sensible heat fluxes. While some of the observed fluxes appear questionable, e.g. both ZM-Mon and SD-Dem appear to lose HFLS seasonality for select periods of their records, in contrast, CLASSIC simulates a more consistent seasonal cycle for all years at these sites. CLASSIC consistently simulates lower latent heat fluxes than evident in





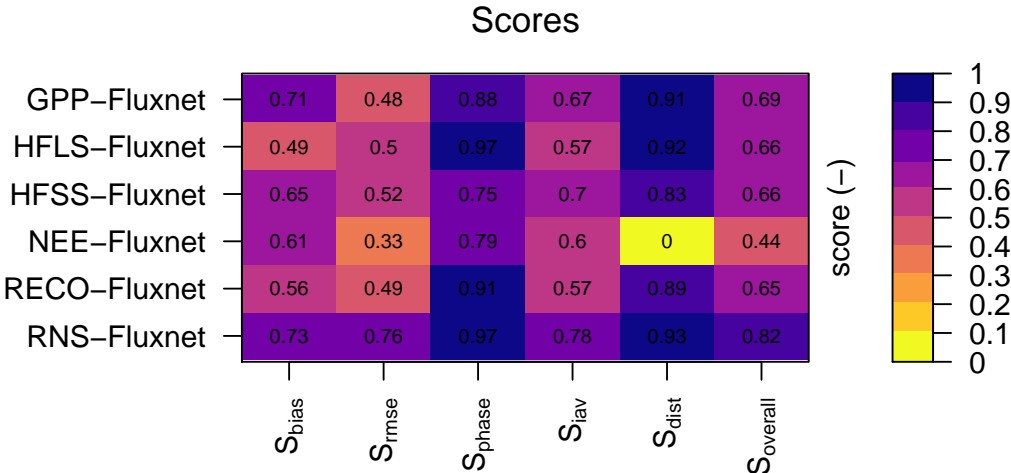

**Figure 4.** AMBER site-level scores comparing CLASSIC to the 31 FLUXNET sites (Table 1). Higher scores indicate better model performance. Further AMBER plots are available in the benchmarking archive listed in Table 2.

the observations. However, non-closure of the energy budget at the FLUXNET sites is a well-known issue. The residual of net radiation plus the ground heat flux minus the latent and sensible heat fluxes ranges from 10 - 30% of RNS across all FLUXNET sites (Wilson et al., 2002; Cui and Chui, 2019). Since CLASSIC enforces closure of the energy budget, this non-closure by the EC data complicates evaluation of simulated energy fluxes (both HFLS and HFSS). For example, the period of highest

mismatch at AU-Tum also corresponds to years with the highest non-closure terms (Fig. 6 and Supplementary Figures S10 and S11) The mean RMSE across sites was 31.5 W m$^{-2}$ with a median R$^2$ of 0.08 (recall the coefficient of determination, as calculated here, can go negative, see Section 5.1. Supplementary Table S2 contains all per site values). The AMBER scores for HFLS were lowest in bias and RMSE (0.49 and 0.50, respectively) giving an overall score of 0.66.

The sensible heat fluxes (HFSS) simulated by CLASSIC were generally closer to the EC tower observations (Figure 7) than

10 HFLS, but with a slight overestimate compared to observations. The mean RMSE across sites was 24.7 W m$^{-2}$ with a median R$^2$ of 0.31 (See Supplementary Table S3 for per site values). The AMBER scores for HFSS were higher than HFLS with the lowest component score being $S_{rmse}$ with 0.52 and an overall score of 0.66, equal to the HFLS overall score.

**5.2.2 Carbon fluxes**

Figure 8 compares the simulated mean seasonal cycle of GPP to FLUXNET observations. While we removed implausible

values from the observations, i.e. negative GPP, we did retain other data points that we suspect could be spurious (e.g. 2005 GPP at SD-Dem in Supplementary Fig. S16). Over all 31 sites, the CLASSIC outputs have an average RMSE of 0.07 kg C m$^{-2}$ month$^{-1}$ and a median R$^2$ of 0.15 (See Supplementary Table 4 for per site values). The AMBER scores for GPP are lowest for $S_{rmse}$ at 0.48 with a $S_{overall}$ of 0.69.





For biomes which are present at multiple sites (EBF, ENF, GRA, DBF), we can draw some general conclusions about the model performance. The EBF sites are primarily located in the low latitudes with the exception of sites in Turkey (GH-Ank) and France (FR-Pue). These mid-latitude EBF sites are for *Quercus ilex* (evergreen Oak) which don't correspond well to the CLASSIC formulation for evergreen broadleaf PFT, which is typically assumed to be a tropical PFT. CLASSIC generally has

a small bias, simulating somewhat lower GPP for the EBF sites (excepting GH-Ank; Fig. 8) with an RMSE for those sites of 0.08 kg C m$^{-2}$ month$^{-1}$. This bias is not evident for the ENF sites where CLASSIC generally simulates the sites well with a mean RMSE of 0.05 kg C m$^{-2}$ month$^{-1}$ but with one Canadian site (CA-Qfo) significantly underestimated. The deciduous broadleaf forest biome sites indicate that CLASSIC may generally underestimate GPP for this biome. The grassland sites are generally well simulated (mean RMSE of 0.5 kg C m$^{-2}$ month$^{-1}$) with the exception of US-Wkg where CLASSIC is unable to

capture the peak GPP values. The Kendall grassland (US-Wkg) is a semi-arid site in southern Arizona, USA. The mean annual rainfall is 345 mm, of which most falls in the summer months (Scott et al., 2010). Due to the extreme moisture stress caused by the low precipitation and heavy drought in 2003 – 2006, CLASSIC is unable to properly establish the vegetation and attain the productivity level observed by the EC tower. This site demonstrates the challenges inherent in this kind of comparison. At US-Wkg, the native vegetation cover of C$_4$ grasses interspersed with C$_3$ shrubs was intact prior to the early 2000s drought.

Post-drought most of the native bunchgrasses were dead, along with high shrub mortality, leading to the establishment of an invasive grass (Scott et al., 2010). The site is also lightly to moderately grazed. Changes in vegetation such as this, along with unique drought conditions (which are only partly captured in the observation period) are difficult to capture adequately by CLASSIC if the goal is to evaluate a model across a large number of sites to capture the diversity of biomes globally. Other sites present similar difficulties to replicate the observed GPP due to complex histories or conditions. As a result it is important

to compare across several sites to reduce the importance of the site-level conditions and to observe the model performance on the whole. The other biomes have too few sites to interpret the results in detail. CLASSIC presently does not represent shrubs as separate PFTs so the shrubland biomes are simulated with trees PFTs in place of shrubs.

Seasonal plots of simulated ecosystem respiration (RECO) are compared against FLUXNET estimated values in Fig. 9. CLASSIC has generally lower variability in RECO than suggested by EC-tower derived estimate,s especially at low latitudes

sites, such as BR-Sa1, GF-Guy, MY-PSO, ZM-Mon, and SD-Dem. In evergreen needleleaf forest sites, the simulated RECO is generally higher than the FLUXNET values. This overestimate is consistent with high GPP values at these sites. Similarly, the low GPP simulated for the evergreen broadleaf forests yields a low biased RECO. MY-PSO is a notable exception with CLASSIC simulating smaller GPP but larger RECO than the EC-tower derived quantities. Also, while RU-SkP (DNF) was reasonably well simulated for GPP, CLASSIC simulates a much larger RECO than the EC-tower derived value. The mean

RMSE across all sites is 0.06 kg C m$^{-2}$ month$^{-1}$ with a median R$^2$ of 0.12 (See Supplementary Table 5 for per site values). The AMBER scores for RECO are similar to the GPP scores with a slightly lower overall score (0.65) but a higher $S_{rmse}$ of 0.49.

NEE is directly observed at eddy-covariance towers. The comparison between model and observations for NEE is, however, confounded due to the issues discussed in Section 5. Due to these factors, the AMBER NEE scores are lower than all others



presented here with an $S_{rmse}$ of 0.33, and $S_{dist}$ of 0, and an overall score of 0.44 (Site-level plots of NEE are included in the supplementary material).

## 6  Future directions for CLASSIC development

CLASSIC remains under active development. Scientific developments include the introduction of non-structural carbohydrates
(NSC) (Asaadi et al., 2018), which addresses a known issue of a delayed spring leaf out for deciduous tree species. The NSC work also lays the groundwork for incorporation of a N-cycle in CLASSIC that is presently in development. Other works in progress includes the incorporation of high-latitude shrub PFTs for both the physical and biogeochemical aspects of this growth form. Following on from recent work demonstrating CLASSIC's capable performance in simulating the physics of permafrost regions (Melton et al., 2019d), CLASSIC's bulk soil C pools will be replaced with an explicit tracking of soil C per soil layer
that is better suited to simulate permafrost C dynamics. As part of this work, a C tracer is being incorporated to allow tracking of $^{14}$C or other isotopes through the model C pools. Planned technical developments include further modularization of the physics code, greater adoption of data structures to move away from lengthy argument lists, and modifications to allow the same code to be seamlessly used offline as well as in the CanESM framework. Benchmarking developments include adding peatland EC tower sites to allow evaluation of our peatland module (Wu et al., 2016) as well as sites for biomes that are
presently poorly represented.

## 7  Conclusions

CLASSIC is the open source community successor to CLASS-CTEM. CLASS and CTEM have decades of development behind them and have been extensively evaluated and applied at scales from site-level, involving evaluation aginst data from EC towers, to coupling to an atmosphere model as part of CanESM. While the science within CLASS-CTEM has continued
to advance in response to changing scientific questions and applications, the technical aspects of CLASS-CTEM have fallen behind. Our study details the transformation of CLASS-CTEM from a primarily internally developed model to one designed to encourage community use and development. Table 2 summarizes the URLs of CLASSIC resources.

The work detailed here to create CLASSIC addresses the barriers to code sharing identified by Easterbrook (2014): portability, configurability, entrenchment, model-data blur, and provenance. Specifically, we have addressed code portability by
providing a software container for CLASSIC to run in serial or parallel modes, along with rewriting the model I/O to use community standard netCDF format files. Configurability of CLASSIC is handled by providing extensive documentation of appropriate model setup via Doxygen, as well as benchmarking sites that can be used as examples of how to configure CLASSIC appropriately along with scripts to properly setup and run CLASSIC over 31 FLUXNET sites. Entrenchment is addressed most easily going forward as we adopt an open community approach to model development including the use of issue trackers
and forums to outline new model developments and bug fixes. Model-data blur is reduced by using the FLUXNET sites for our site-level benchmarking as they required minimal processing prior to use. Lastly, provenance is made possible by our use



of a Singularity software container and our benchmarking pipeline (Appendix A) where any user can reproduce the site-level plots and AMBER scores presented here. AMBER is made available to the community as a means to guide model development where low scores highlight possible areas for improvement. Additionally the use of AMBER can help ensure that improvements in one area of CLASSIC are not degrading the performance of other variables. We encourage members of the scientific community to use, develop, and contribute back to CLASSIC.

*Code and data availability.* The CLASSIC software container, CLASSIC site-level FLUXNET benchmarking data and AMBER reports, and all code both for CLASSIC v. 1.0 and for preparing the plots of model outputs as well as site-level data presented here are archived on the CLASSIC community Zenodo page. (Melton et al., 2019a, b, c). See Table 2 for the location of all resources.





**Table 1.** FLUXNET2015 sites used in the FLUXNET site-level benchmarking (Section 5.2).

| Site ID | Site name | Latitude | Longitude | Elevation (m) | Years | IGBF code[a] | DOI |
|---|---|---|---|---|---|---|---|
| AU-Tum | Tumbarumba | -35.6566 | 148.1517 | 1200 | 2001 - 2013 | EBF | 10.18140/FLX/1440126 |
| BR-Sa1 | Santarem-Km67-Primary Forest | -2.8567 | -54.9589 | 88 | 2002 - 2011 | EBF | 10.18140/FLX/1440032 |
| CA-Qfo | Quebec - E. Boreal, Mature Black Spruce | 49.6925 | -74.3421 | 382 | 2003 - 2010 | ENF | 10.18140/FLX/1440045 |
| CA-TPD | Ontario - Turkey Point Mature Deciduous | 42.6353 | -80.5577 | 260 | 2012 - 2014 | DBF | 10.18140/FLX/1440112 |
| CG-Tch | Tchizalamou | -4.2892 | 11.6564 | 82 | 2006 - 2009 | SAV | 10.18140/FLX/1440142 |
| CN-Dan | Dangxiong | 30.4978 | 91.0664 | 4313 | 2004 - 2005 | GRA | 10.18140/FLX/1440138 |
| CZ-BK1 | Bily Kriz forest | 49.5021 | 18.5369 | 875 | 2004 - 2014 | ENF | 10.18140/FLX/1440143 |
| DE-Kli | Klingenberg | 50.8931 | 13.5224 | 478 | 2004 - 2014 | CRO | 10.18140/FLX/1440149 |
| DE-Tha | Tharandt | 50.9624 | 13.5652 | 385 | 1996 - 2014 | ENF | 10.18140/FLX/1440152 |
| DK-Sor | Soroe | 55.4859 | 11.6446 | 40 | 1996 - 2014 | DBF | 10.18140/FLX/1440155 |
| FI-Hyy | Hyytiala | 61.8474 | 24.2948 | 181 | 1996 - 2014 | ENF | 10.18140/FLX/1440158 |
| FR-Fon | Fontainebleau-Barbeau | 48.4764 | 2.7801 | 103 | 2005 - 2014 | DBF | 10.18140/FLX/1440161 |
| FR-Pue | Puechabon | 43.7413 | 3.5957 | 270 | 2000 - 2014 | EBF | 10.18140/FLX/1440164 |
| ES-LgS | Laguna Seca | 37.0979 | -2.9658 | 2267 | 2007 - 2009 | OSH | 10.18140/FLX/1440225 |
| GF-Guy | Guyaflux (French Guiana) | 5.2788 | -52.9249 | 48 | 2004 - 2014 | EBF | 10.18140/FLX/1440165 |
| GH-Ank | Ankasa | 5.2685 | -2.6942 | 124 | 2011 - 2014 | EBF | 10.18140/FLX/1440229 |
| IT-Lav | Lavarone | 45.9562 | 11.2813 | 1353 | 2003 - 2014 | ENF | 10.18140/FLX/1440169 |
| IT-SRo | San Rossore | 43.7279 | 10.2844 | 6 | 1999 - 2012 | ENF | 10.18140/FLX/1440176 |
| IT-Tor | Torgnon | 45.8444 | 7.5781 | 2160 | 1999 - 2012 | GRA | 10.18140/FLX/1440237 |
| MY-PSO | Pasoh Forest Reserve | 2.9730 | 102.3062 | 147 | 2003 - 2009 | EBF | 10.18140/FLX/1440240 |
| NL-Loo | Loobos | 52.1666 | 5.7436 | 25 | 1996 - 2014 | ENF | 10.18140/FLX/1440178 |
| PA-SPs | Sardinilla-Pasture | 9.3138 | -79.6314 | 68 | 2007 - 2009 | GRA | 10.18140/FLX/1440179 |
| RU-Che | Cherski | 68.6130 | 161.3414 | 6 | 2002 - 2005 | WET | 10.18140/FLX/1440181 |
| RU-Fyo | Fyodorovskoye | 56.4615 | 32.9221 | 265 | 1998 - 2014 | ENF | 10.18140/FLX/1440183 |
| RU-Ha1 | Hakasia steppe | 54.7252 | 90.0022 | 446 | 2002 - 2004 | GRA | 10.18140/FLX/1440184 |
| RU-SkP | Yakutsk Spasskaya Pad larch | 62.2550 | 129.1680 | 246 | 2012 - 2014 | DNF | 10.18140/FLX/1440243 |
| SD-Dem | Demokeya | 13.2829 | 30.4783 | 500 | 2005 - 2009 | SAV | 10.18140/FLX/1440186 |
| US-WCr | Willow Creek | 45.8059 | -90.0799 | 520 | 1999 - 2014 | DBF | 10.18140/FLX/1440095 |
| US-Wkg | Walnut Gulch Kendall Grasslands | 31.7365 | -109.9419 | 1531 | 2004 - 2014 | GRA | 10.18140/FLX/1440096 |
| ZA-Kru | Skukuza | -25.0197 | 31.4969 | 359 | 2000 - 2013 | SAV | 10.18140/FLX/1440188 |
| ZM-Mon | Mongu | -15.4378 | 23.2528 | 1053 | 2007 - 2009 | DBF | 10.18140/FLX/1440189 |

International Geosphere-Biosphere Programme (IGBP) land classification abbreviations used include Evergreen Broadleaf Forests (EBF), Evergreen Needleleaf Forests (ENF), Savannas (SAV), Deciduous Broadleaf Forests (DBF), Grasslands (GRA), Croplands (CRO), Open Shrublands (OSH), Closed Shrublands (CSH), Permanent Wetlands (WET), and Deciduous Needleleaf Forests (DNF).





**Figure 5.** Mean seasonal monthly net radiation for the 31 FLUXNET sites and CLASSIC simulated values. Shaded regions indicate the standard deviation across the sample years. The years of observations for each site as listed along with the site name and IGBF biome which correspond to evergreen broadleaf forest, EBF; evergreen needleleaf forest, ENF: grasslands, GRA; deciduous broadleaf forest, DBF; croplands, CRO; open shrublands, OSH; permanent wetlands, WET; and deciduous broadleaf forests, DBF.



**Figure 6.** Monthly observed and simulated latent heat fluxes for 23 FLUXNET sites. See caption of Figure 5 for biome names. Shaded regions indicate the standard deviation across the sample years. Table 1 lists the full site names.





**Figure 7.** Monthly observed and simulated latent heat fluxes for 23 FLUXNET sites. See caption of Figure 5 for biome names. Shaded regions indicate the standard deviation across the sample years. Table 1 lists the full site names.



**Figure 8.** The mean seasonal cycle of GPP for 31 FLUXNET sites as simulated by CLASSIC. IGBP biome of each site is listed alongside its FLUXNET site ID (see Table 1). Shaded regions indicate the standard deviation across the sample years. See caption of Figure 5 for biome names.



**Figure 9.** The mean seasonal cycle of RECO for 31 FLUXNET sites as simulated by CLASSIC. IGBP biome of each site is listed alongside its FLUXNET name (see Table 1). Shaded regions indicate the standard deviation across the sample years. See caption of Figure 5 for biome names.



**Table 2.** Summary table of CLASSIC Resources. Resources that do not list a version number will be updated with the most recent versions through time.

| | |
|---|---|
| CLASSIC webpage | https://cccma.gitlab.io/classic |
| CLASSIC Gitlab code repository | https://gitlab.com/cccma/classic |
| CLASSIC Community Zenodo page | https://zenodo.org/communities/classic/ |
| CLASSIC code for v. 1.0 | https://zenodo.org/record/3522407 |
| CLASSIC Singularity container v. 1.0 (including AMBER v. 0.1.5) | https://zenodo.org/record/3525249 |
| Benchmarking data and outputs for CLASSIC v. 1.0 (including those generated by AMBER) | https://zenodo.org/record/3525336 |
| AMBER CRAN page | https://cran.r-project.org/package=amber |
| CLASSIC discussion forum | http://classic-message-board.158658.n8.nabble.com/ |





**Appendix A: Quick start guide to running CLASSIC at FLUXNET sites**

This document will walk through the basic setup, compilation, and running of CLASSIC at FLUXNET sites.

**Requirements**

This guide assumes that the reader is using a Linux machine, and that they have Singularity and tar installed. The installa-
tion guide for Singularity can be found at https://sylabs.io/docs/#singularity. Tar can be installed from https://www.gnu.org/
software/tar/. If using a Windows or Mac machine see instructions on setting up Singularity at https://sylabs.io/guides/3.3/
user-guide/installation.html#install-on-windows-or-mac.

**Obtaining the Source Code**

The first step is setting up a directory structure. For the sake of this guide, we will work out of a directory located at,
'/home/eccc'. So first we navigate to our chosen directory location:

```
cd /home/eccc
```

The CLASSIC source code is hosted on GitLab, and can be cloned with:

```
git clone https://gitlab.com/cccma/CLASSIC.git
```

If you are familiar with git's workflow and wish to contribute to the codebase in the future, then it is a good idea to first fork
a copy to your own GitLab account. Once you have a fork, you can clone the repository with:

```
git clone https://gitlab.com/**your_gitlab_username**/CLASSIC.git
```

Once the cloning process is complete, there should be a directory titled CLASSIC located in your directory:

```
/home/eccc/CLASSIC
```

**Obtaining other necessary files**

All other necessary files are found on Zenodo in the form of compressed packages. We will use the 'tar' command to un-
pack them beside the CLASSIC repository. First, navigate to the CLASSIC community Zenodo page at https://zenodo.org/
communities/classic and download the FLUXNET, and CLASSIC_container tar files. Your directory should now contain the
following:

```
/home/eccc/CLASSIC
/home/eccc/CLASSIC_container.tar.gz
/home/eccc/FLUXNET.tar.gz
```

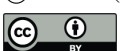



From the '/home/eccc' directory, we will decompress the tarballs and move the decompressed files to their correct locations. The following list of commands will accomplish this:

```
tar xzvf CLASSIC_container.tar.gz
tar xzvf FLUXNET.tar.gz
```

```
mv CLASSIC_container.simg CLASSIC/

mv FLUXNET/TRENDY_CO2_1700_2018.nc CLASSIC/inputFiles/
mv FLUXNET/FLUXNETsites CLASSIC/inputFiles/
```
```
mv FLUXNET/observationalDataFLUXNET CLASSIC/inputFiles/
mv FLUXNET/benchmark_CLASSIC_output CLASSIC/benchmark/
mv FLUXNET/benchmark_CLASSIC_plots CLASSIC/benchmark/
mv FLUXNET/benchmark_CLASSIC_AMBER CLASSIC/benchmark/
rm -rf FLUXNET/
```

**Using the container**

Navigate to the CLASSIC root directory. The rest of the tutorial will assume that you stay in this working directory:

```
cd /home/eccc/CLASSIC
```

Since much of CLASSIC's functionality relies on the libraries contained within the container, we execute commands through the container with this syntax:

```
singularity exec CLASSIC_container.simg [commands here]
```

Most of the calls like this are contained within the automated run scripts so their direct use is minimized. However, if you wish to deviate from the automated run scripts, that command will be of great use.

First, we'll make sure we have a clean working directory.

```
singularity exec CLASSIC_container.simg make mode=serial clean
```

We are now ready to compile the source code for serial simulation. This is done with the command:

```
singularity exec CLASSIC_container.simg make mode=serial
```

The compilation process can take several minutes. If you get errors, check that you have the latest stable version of CLASSIC pulled from the repository.





**Setting up FLUXNET site-level runs**

With the FLUXNET sites in place and CLASSIC compiled, we can now setup the job options file(s) to run over the FLUXNET sites. To do this, run the job options script by invoking:

```
tools/siteLevelFLUXNET/prep_jobopts.sh
```

This script puts a new job-options file in every FLUXNET site directory tailored to that particular site. The 'singularity exec' command is not necessary for this particular script.

**Running CLASSIC over FLUXNET sites**

This guide will cover two methods for running the CLASSIC binary on FLUXNET sites. They can be run individually, or through the batch-run script.

**Running all sites with the batch script (recommended)**

A script is provided in the CLASSIC repository to run all FLUXNET sites, provided the directory structure of this document is followed. The script will run for all sites it finds within the inputFiles/FLUXNETsites directory and is invoked by:

```
tools/siteLevelFLUXNET/run_sites.sh
```

    All model restart files are provided in a 'pre-spunup' state. Still, running all sites can a lengthy process on older machines,
and sometimes you may not want or need all of them. To reduce the number of sites being run, remove the undesired site directories from inputFiles/FLUXNETsites. For example, if you don't need to process 'DK-Sor', you could do:

```
mv inputFiles/FLUXNETsites/DK-Sor inputFiles
```

    This would simply move the site out of the way, and it could be placed back into the FLUXNETsites directory when needed.

**Running individual sites (advanced)**

Individual sites are run through the Singularity container directly referencing the job options file for that particular site. The command takes the form:

```
singularity exec [Location of CLASSIC container] [location of CLASSIC binary]\
 [Location of job options file for that site] [Longitude/Latitude of the site]
```

    Since this is a site-level simulation the shorthand 0/0 or 0/0/0/0 can be used in place of the actual longitude and latitude of
the site. Piecing it all together, if we want to run on the site AU-Tum, the command would be:

```
singularity exec CLASSIC_container.simg bin/CLASSIC_serial \
 inputFiles/FLUXNETsites/AU-Tum/job_options_file.txt 0/0/0/0
```





If you decide to shell into the singularity container (recommended if you're familiar with Singularity and terminal commands), then this becomes

```
bin/CLASSIC_serial inputFiles/FLUXNETsites/AU-Tum/job_options_file.txt 0/0/0/0
```

More information on running CLASSIC can be found in the CLASSIC manual at https://cccma.gitlab.io/classic/index.html.

## 5  Processing output

Whether you ran multiple sites with the batch job, or a single site individually, the output can be found in 'outputFiles/FLUXNETsites/sitename/'. These outputs are in the form of netCDF files. More information on netCDF can be found at https://www.unidata.ucar.edu/software/netcdf/, but briefly, it is a machine-agnostic array-oriented form of data storage.

A script is provided to convert this data to csv format, which may be easier to work with if unfamilier with netCDFs, as well as generate plots of several output variables against observational values of those variables. The script is run with:

```
tools/siteLevelFLUXNET/process_outputs.sh
```

**NOTE**: if not all sites were run with the batch script, errors will be seen in the output of this script. This is expected, and does not mean that the processing has failed.

Once the output plots have been generated, the 'process_outputs' script will run the Automated Model Benchmarking (AMBER) tool. More information on AMBER can be found at https://cran.rstudio.com/web/packages/amber/amber.pdf.

After the script completes, PDF copies of the plots are found in 'outputFiles/plots', while AMBER results are in 'outputFiles/AMBER'.





**Table A1.** Equations used for computing spatially integrated scalar scores for each variable. The four steps refer to (1) the definition of the statistical metric, (2) nondimensionalization, (3) conversion to unit interval, and (4) spatial integration. The integration limits $t_0$ and $t_f$ are the initial and final time step, respectively. The variables $c_{mod}$ and $c_{ref}$ refer to the monthly mean annual cycle of the model and reference data, respectively. $crmse$ is the 'centralized RMSE', which is the RMSE of the anomalies. This ensures that the RMSE score excludes also considering the bias, which is captured by the bias score.

| Score | Step | Equation |
|---|---|---|
| $S_{bias}$ | (1) | $bias(\lambda,\phi) = \overline{v_{mod}}(\lambda,\phi) - \overline{v_{ref}}(\lambda,\phi)$ |
| | (2) | $\varepsilon_{bias} = |bias(\lambda,\phi)|/\sigma_{ref}(\lambda,\phi)$, where $\sigma_{ref}$ is the standard deviation of the reference data |
| | (3) | $s_{bias}(\lambda,\phi) = e^{-\varepsilon_{bias}(\lambda,\phi)}$ |
| | (4) | $S_{bias} = \overline{\overline{s_{bias}}}$ |
| $S_{rmse}$ | (1) | $rmse(\lambda,\phi) = \sqrt{\frac{1}{t_f - t_0} \int_{t_0}^{t_f} (v_{mod}(t,\lambda,\phi) - v_{ref}(t,\lambda,\phi))^2 dt}$ |
| | (2) | $\varepsilon_{rmse}(\lambda,\phi) = crmse(\lambda,\phi)/\sigma_{ref}(\lambda,\phi)$, where $crmse(\lambda,\phi)$ |
| | | $= \sqrt{\frac{1}{t_f - t_0} \int_{t_0}^{t_f} [(v_{mod}(t,\lambda,\phi) - \overline{v_{mod}}(\lambda,\phi)) - (v_{ref}(t,\lambda,\phi) - \overline{v_{ref}}(\lambda,\phi))]^2 dt}$ |
| | (3) | $s_{rmse}(\lambda,\phi) = e^{-\varepsilon_{rmse}(\lambda,\phi)}$ |
| | (4) | $S_{rmse} = \overline{\overline{s_{rmse}}}$ |
| $S_{phase}$ | (1) | $\theta(\lambda,\phi) = \max(c_{mod}(t,\lambda,\phi)) - \max(c_{ref}(t,\lambda,\phi))$ |
| | (2) | not applicable, as units are consistent across all variables |
| | (3) | $s_{phase}(\lambda,\phi) = \frac{1}{2}[1 + \cos(\frac{2\pi\theta(\lambda,\phi)}{365})]$ |
| | (4) | $S_{phase} = \overline{\overline{s_{phase}}}$ |
| $S_{iav}$ | (1) | $iav_{ref}(\lambda,\phi) = \sqrt{\frac{1}{t_f - t_0} \int_{t_0}^{t_f} (v_{ref}(t,\lambda,\phi) - c_{ref}(t,\lambda,\phi))^2 dt}$, |
| | | $iav_{mod}(\lambda,\phi) = \sqrt{\frac{1}{t_f - t_0} \int_{t_0}^{t_f} (v_{mod}(t,\lambda,\phi) - c_{mod}(t,\lambda,\phi))^2 dt}$ |
| | (2) | $\varepsilon_{iav} = |(iav_{mod}(\lambda,\phi) - iav_{ref}(\lambda,\phi))|/iav_{ref}(\lambda,\phi)$ |
| | (3) | $s_{iav}(\lambda,\phi) = e^{-\varepsilon_{iav}(\lambda,\phi)}$ |
| | (4) | $S_{iav} = \overline{\overline{s_{iav}}}$ |
| $S_{dist}$ | (1) | $\sigma = \sigma_{\overline{v_{mod}}}/\sigma_{\overline{v_{ref}}}$ |
| | (2) | not applicable |
| | (3) | not applicable |
| | (4) | $S_{dist} = 2(1 + R)/(\sigma + \frac{1}{\sigma})^2$, |
| | | where $R$ is the spatial correlation coefficient of $\overline{v_{ref}}(\lambda,\phi)$ and $\overline{v_{mod}}(\lambda,\phi)$ |





**Table A2.** Site-level statistical metrics for calculating $S_{rmse}$ at the 31 FLUXNET sites. $crmse$ is the 'centralized RMSE', which is the RMSE of the anomalies. This ensures that the RMSE score excludes also considering the bias, which is captured by the bias score.

| variable | unit | $rmse$ | $crmse$ | $\sigma_{ref}$ | $\epsilon_{rmse}$ (-) | $S_{rmse}$ (-) |
|---|---|---|---|---|---|---|
| GPP | gC m$^{-2}$ day$^{-1}$ | 2.26 | 1.88 | 2.74 | 0.78 | 0.48 |
| HFLS | W m$^{-2}$ | 32.37 | 22.31 | 31.50 | 0.72 | 0.50 |
| HFSS | W m$^{-2}$ | 25.10 | 19.55 | 32.03 | 0.70 | 0.52 |
| NEE | gC m$^{-2}$ day$^{-1}$ | 1.95 | 1.56 | 1.48 | 1.24 | 0.33 |
| RECO | gC m$^{-2}$ day$^{-1}$ | 1.90 | 1.29 | 1.82 | 0.75 | 0.49 |
| RNS | W m$^{-2}$ | 19.34 | 12.32 | 48.56 | 0.28 | 0.76 |



*Author contributions.* J.M. lead the conversion of CLASS-CTEM to CLASSIC, coordinated activities, and wrote the first draft of the manuscript. V. A. created CTEM, originally suggested the possibility of converting CLASS-CTEM to a community model, and contributed to the planning and implementation of CLASSIC along with contributing text. C. S. wrote AMBER, provided AMBER outputs and wrote sections of the manuscript. E. W-C wrote CLASSIC code around netCDF I/O, the MPI incorporation, the meteorological disaggregation,

and the XML editor and also contributed text to the manuscript. E.C. worked on the model compilation scripts, aspects of CLASSIC code, and its use on supercomputers. L. T. processed FLUXNET data and setup the model initialization files for the tower sites used in the site-level evaluation. M.F. wrote and applied the linter used to refactor the code, drafted the coding standards documents, developed the parallel container recipe, and wrote the plotting scripts. All authors contributed to the final manuscript.

*Competing interests.* The authors declare no competing interests

*Acknowledgements.* The authors wish to acknowledge the meticulous work of Diana Verseghy who lead development of the CLASS model from its origins until her retirement in January, 2017. We thank C. Le Quéré for allowing us to distribute her $CO_2$ record that was originally made for the TRENDY project. L.T.'s visit to CCCma was funded by Hamburglobal.

This work used eddy covariance data acquired and shared by the FLUXNET community, including these networks: AmeriFlux, AfriFlux, AsiaFlux, CarboAfrica, CarboEuropeIP, CarboItaly, CarboMont, ChinaFlux, Fluxnet-Canada, GreenGrass, ICOS, KoFlux, LBA, NECC,

OzFlux-TERN, TCOS-Siberia, and USCCC. The ERA-Interim reanalysis data are provided by ECMWF and processed by LSCE. The FLUXNET eddy covariance data processing and harmonization was carried out by the European Fluxes Database Cluster, AmeriFlux Management Project, and Fluxdata project of FLUXNET, with the support of CDIAC and ICOS Ecosystem Thematic Center, and the OzFlux, ChinaFlux and AsiaFlux offices.





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
