# Peer review of "CLASSIC v1.0: the open-source community successor to the Canadian Land Surface Scheme (CLASS) and the Canadian Terrestrial Ecosystem Model (CTEM) - Part 1: Model framework and site-level performance"

_Geoscientific Model Development, 2019_

## Referee Comment (RC1) · Anonymous Referee #1 · 9 Jan 2020

This manuscript describes the CLASSIC v1.0 model, which is a combination of the CLASS and CTEM models. In particular CLASSIC represents a move to open-source development, with additional features to facilitate the use of the model by a diverse community in a range of settings. The paper also gives an overview of the performance of the model at a number of sites.

Overall the paper was well written and clear, and I consider it would be suitable for

publication after minor corrections.

This will be a relatively short review as I didn't find myself with many questions - essentially the paper appears to be a reasonable summary of essentially 'technical' developments to the modelling system. Many of my comments below are suggestions, which of course the authors might wish to argue against.

The only major thing that I think is missing is some discussion about governance - how will the community be run and governed so that the core model develops in a way that satisfies all or a majority of users? How will the trunk code be managed? How will the potentially conflicting needs of different user groups be managed? For example, I assume there are key stakeholders, such as the climate and earth system modellers who need a reliable, highly-optimised code, but equally there might be people who want to use CLASSIC to study a particular process that is perhaps only important in a small geographical area and/or to a few people but which requires a bigger change to the code base. That might be a rather extreme contrast, but my point is that I would like to know about the proposed process by which the main model trunk is managed - who is in control?

Slightly less important, but recurring and so I will mention it here: On P5 and elsewhere it wasn't clear to me whether CLASSIC supports the use of CLASS without CTEM, i.e. can one prescribe vegetation characteristics (such as height and LAI) rather than having these simulated by CTEM? I suspect this configuration is possible, but it is unclear. Such a configuration arguably makes some evaluations (such as against FLUXNET sites) more meaningful - e.g. looking at the turbulent fluxes is less interesting if CTEM has decided to produce vegetation that is very different from reality at a site (which also relates to the difficulties discussed in spinning up long-timescale processes with only a few years of meteorological data). And if this functionality does exist in CLASSIC, why not employ it in (at least some of) the FLUXNET-based evaluation described in Section 5? (I appreciate that prescribing vegetation characteristics is not a panacea for all of the difficulties outlines in Sec.5.)

**Specific comments**

P2 L15 Benefits of OSS: I wonder if the creation of a community of collaborating modellers might be considered another benefit of the OSS/community approach (though arguably it might come under your first point 'affordability').

Also on L10 you appear to equate OSS development with being a community model, but these could be considered separate - OSS means the code is accessible to all, but a community model *might* imply more than that in terms of there being a coordinated approach across groups to develop and use the model. I think it is this second sense that you intend, consistent with the aims of the other models you cite (JULES, CLM, etc.). Perhaps you could expand upon some of these aspects.

I'm not convinced that L16 'flexibility of configuration' is a benefit from OSS. Flexibility is a desirable characteristic of the model, but by itself OSS does not give this (but flexibility can become more affordable if OSS provides a bigger community of developers). I think you should remove this from the benefits section - or at least rephrase/recast the idea.

P1 L29 key features: You later describe how CLASSIC provides a means to check that results are reproducable (checksums etc.), which could just about fall under your 8th point here (benchmarking). To me this is an important aspect in its own right and might deserve explicit mention here.

P3 L30 'standard offline model setup' - what is this and where can a user find it (and other configurations)? Later you mention that various runs can be set up and performed using the information provided as part of the package/container, but in general I was unclear as to where (if at all) standard configurations can be obtained. Version-controlled code and the container environment are important, but obviously the model set up is also critical. At some point in the manuscript I would like to see some (or clearer) coverage of where standard configurations can be acquired. On P3 it might suffice to point to this information, e.g. "see Section X".

P9 L7-18: The level of detail and discussion about checksums is unnecessary. The basic principles are well-established and can (most likely) be cited. Most or all of these lines can be removed.

P9 Sec 3.2.3: I had a quick look at some of the material on Zenodo and it is possible that I didn't find the correct material, but I am wondering if there is any collated scientific documentation for CLASSIC. (In fact I'm thinking of science equations rather than the more code-specific documentation that is the focus of this section. I suspect the answer is 'no', as is the case for many models. Section 2 outlines the science and provides various references that can be followed up, but for many models it takes a keen student and days of work to assemble the definitive description of the science in the latest version of the model. As far as I can see the self-documenting code is very useful but does not address the more detailed scientific documentation. Are there any plans for further documentation? As I said, many models face similar challenges, so this is not a particular criticism of CLASSIC alone!)

P10 L25-end: The examples of how to select domains are unnecessary detail in such an overview paper; it is sufficient to know that the capability exists. I suggest removing this.

P11 L7-8: I don't understand this. It reads as if FORTRAN is used to convert namelist files into netCDF. Perhaps the ASCII files listed in the namelists are covered into netCDF? If so, rephrase.

P13 L22: I'm only moderately familiar with ILAMB but I thought it mainly (or solely) used colours to show rankings of results (rather than whether thresholds are passed). Also, even more pedantically, I think ILAMB and other tools can be very useful when assessing and presenting how different versions or configurations of a single model compare - in which case your objection to its use for a single model (L23) would not hold.

P6 L13: Being able to use the same code in standalone and coupled applications is,

of course, highly advantageous. I know other models also struggle to ensure that a single code base can be used for both (with the main difficulties often being at the higher-level interfaces to parent structures). To me this is an important consideration that could (possibly) be brought up earlier in the manuscript (perhaps as an important aspiration for CLASSIC, even if in fact it is still in the future). As it is, some readers might be slightly confused by the earlier claims that the models are used in coupled applications, yet at this late stage they are told the codes differ (even if in fact the differences are largely 'technical' rather than 'scientific').

**Minor comments**

P3 L10 'second generation' - I never like these terms as they are so vague, but it's just about OK as you quote it! I can't recall who first coined the term (Sellers, Shuttleworth?) but perhaps you could consider a citation (or explanation) to back it up.

P5 L14 "nine PFTs that map directly onto...": Is it that the 9 PFTs of CTEM are collapsed onto the 4 PFTs used by CLASS, so that two or more CTEM PFTs are represented by one CLASS PFT? This could be made clearer. I assume there is some motivating evidence that supports that this is a reasonable approach that saves CPU time.

P5 L28 fire: I realise Fig.1 is just a schematic, but possibly some representation of fire (and land use) could be included.

P8 I found the order of some sections in this area was slightly unexpected. Looking back, perhaps it was mainly that I thought the containerisation section might come later, after code design etc., as it is essentially a way to bundle everything that has been created/described previously. Coding standards and docuemenation I would also push up the list. The move to Git/Gitlab I would describe before other details such as how to add new PFTs. Similarly, P9 L20 mentions the move away from fixed-form FORTRAN...but mentally I have already checked outputs using checksums in the previous section. To me anything about the code repository and coding standards comes

before checksums. But really this is personal preference, and your order is fine!

P10 L20: As I read this, point runs can produce gridded (2-D) outputs. Is that really the case?

P11 L22 and others: I'd prefer "3 or 6" rather than "3/6" which I keep reading as a fraction. I assume CLASSIC is fairly flexible in terms of timestep lengths and data intervals, but this section (and maybe others) does read a bit as if only certain timesteps are allowed. e.g. L21 says inputs are typically 3 or 6 hourly (which is fine), but later in the paragraph we are talking about "3/6" hourly data and a 30 minute timestep - i.e. it becomes specific. Consider clarifying that the model is indeed more flexible than this.

P12 Sec4.2: Consider clarifying at this stage that AMBER is included in the container (as is later made clear).

**Typos etc.**

P3 L31: add a comma after 61m.

P10 L20: 'also' seems to be misplaced here. Remove?

P11 L4: remove 'which'

P12 L12: remove 'and'

P13 L21 (and at least one other location): change "don't" to "do not"

P14 L1 'as well': I'd prefer 'additionally'.

P15 L11: change 'Both' to 'both'

P18 L24: Misplaced comma (and I'd remove the first comma on L25 too).

P39 L38: I haven't pored over all the references, but by chance I noticed a slightly dubious entry here: '0, null'.

---

## Referee Comment (RC2) · Anonymous Referee #2 · 16 Apr 2020

Moving CLASS-CTEM to an open source community model is an important initiative. This paper is a well-written discussion – perhaps a little unbalanced in the level of technical detail between different sections and with a few typos to pick up in copy editing, but those are not major problems. I judge that is acceptable for publication essentially as it stands, but I have some minor suggestions and questions:

- soil freezing and fire are mentioned in the text but not represented in Figure 1.

[Figure]

- can separate reference heights be provided for temperature and wind speed? They often differ, both in site measurements and global datasets

- total precipitation is required as an input, but what flexibility is there in specifying how the model will divide it into snow and rain?

- the single model snow layer is a historical feature of CLASS, but is there a scientific justification for maintaining it in a model with many more ground layers?

- other models have used FLUXNET data for benchmarking. Could a short discussion of CLASSIC performance in the context of other studies be added?

———————————————

---

## Author Comment (AC1) · 30 Apr 2020

**Author reply**

Dear Gerd Folberth,

We wish to thank our two reviewers for their time and considered comments on our manuscript. Below we lay out their comments in full and include our replies in bold font.

**Anonymous Referee #1**

This manuscript describes the CLASSIC v1.0 model, which is a combination of the CLASS and CTEM models. In particular CLASSIC represents a move to open-source development, with additional features to facilitate the use of the model by a diverse community in a range of settings. The paper also gives an overview of the performance of the model at a number of sites.

Overall the paper was well written and clear, and I consider it would be suitable for publication after minor corrections.

This will be a relatively short review as I didn't find myself with many questions - essentially the paper appears to be a reasonable summary of essentially 'technical' developments to the modelling system. Many of my comments below are suggestions, which of course the authors might wish to argue against.

**Thank you for your comments. We are glad to hear of your generally positive view of our work.**

The only major thing that I think is missing is some discussion about governance - how will the community be run and governed so that the core model develops in a way that satisfies all or a majority of users? How will the trunk code be managed? How will the potentially conflicting needs of different user groups be managed? For example, I assume there are key stakeholders, such as the climate and earth system modellers who need a reliable, highly-optimised code, but equally there might be people who want to use CLASSIC to study a particular process that is perhaps only important in a small geographical area and/or to a few people but which requires a bigger change to the code base. That might be a rather extreme contrast, but my point is that I would like to know about the proposed process by which the main model trunk is managed - who is in control?

**This is an interesting question. While we see some value in spelling out the model governance early, we feel the role of this paper is to focus on the technical and scientific transition of CLASS-CTEM to a community model format. As well, ideally, the model governance structure will be developed in partnership with the CLASSIC community. All of the questions listed above are likely to be ones that we will need to answer over time. While CLASSIC's primary use, historically, has been the CanESM family of models, it also has significant applications in other regional climate models, hydrologic modelling, and site-level studies. It would disingenuous to claim to adopt a community approach and then decide a priori which model purpose is most important. That said, internal to ECCC, the model application will be tightly linked to the development of CanESM and its influence on CanESM performance will be closely scrutinized. In the end, decisions around model parameterization adoption are likely to be relatively simple. As we have been developing an extensive benchmarking apparatus for CLASSIC (AMBER), model development decisions will be tightly linked to model benchmarking results. Changes will be adopted if they lead to demonstrable improvements in model performance. This approach also appears to be the experience with CLM development (D. Lawrence, pers. comm. 2020).**

Slightly less important, but recurring and so I will mention it here: On P5 and elsewhere it wasn't clear to me whether CLASSIC supports the use of CLASS without CTEM, i.e. can one prescribe vegetation characteristics (such as height and LAI) rather than having these simulated by CTEM? I suspect this configuration is possible, but it is unclear. Such a configuration arguably makes some evaluations (such as against FLUXNET sites) more meaningful - e.g. looking at the turbulent fluxes is less interesting if CTEM has decided to produce vegetation that is very different from reality at a site (which also relates to the difficulties discussed in spinning up long-timescale processes with only a few years of meteorological data). And if this functionality does exist in CLASSIC, why not employ it in (at least some of) the FLUXNET-based evaluation described in Section 5? (I appreciate that prescribing vegetation characteristics is not a panacea for all of the difficulties outlines in Sec.5.)

Yes indeed, CLASSIC still retains the ability to turn off the biogeochemical cycling (CTEM). We have now added an explicit mention of the ability to turn off the biogeochemistry to Section 2.1, "While CLASSIC includes a biogeochemical component (CTEM; described below), it is possible to turn it off and run CLASSIC with specified vegetation structural attibutes such as tree height, plant area index, etc, which is desirable in some situations. ". We didn't use this capability in this study as we are primarily interested in benchmarking how the model performs as a whole, i.e. including both the physics and biogeochemistry. The ability of the model to spinup the C pools (without the issue of limited years of meteorology) is investigated in a companion paper that will look at CLASSIC's global scale performance.

**Specific comments**

P2 L15 Benefits of OSS: I wonder if the creation of a community of collaborating modellers might be considered another benefit of the OSS/community approach (though arguably it might come under your first point 'affordability').

**We feel that is a benefit from the community aspect, and not necessarily an OSS benefit. This links into our answer to the comment below.**

Also on L10 you appear to equate OSS development with being a community model, but these could be considered separate - OSS means the code is accessible to all, but a community model might imply more than that in terms of there being a coordinated approach across groups to develop and use the model. I think it is this second sense that you intend, consistent with the aims of the other models you cite (JULES, CLM, etc.). Perhaps you could expand upon some of these aspects.

**Yes, we didn't wish to imply OSS and a community model are one and the same. Any model that has put their code in a public repository could argue that they are OSS. To be a community model though, requires a stronger connection between collaborating groups as well as a sense of ownership over the model and its direction. To better convey that sentiment we have reword the beginning of the paragraph from:**

**"OSS development in land surface modelling presents several benefits to its participants including: 1) affordability, creating a new land surface scheme is a massive undertaking; 2) transparency, as the code is open to full scrutiny; 3) flexibility, the models are designed to be both used in their present configuration and also extended to answer new science questions; and 4) perpetuity, many users across diverse institutions help protect the code against loss, deletion, or obsolescence. However…"**

**to**

**"A community approach to land surface modelling presents several benefits to its participants … new science questions; 4) perpetuity, many users across diverse institutions help protect the code against loss, deletion, or obsolescence; and 5) a collaborative community invested in developing, applying, and improving a common resource - the model. However…"**

I'm not convinced that L16 'flexibility of configuration' is a benefit from OSS. Flexibility is a desirable characteristic of the model, but by itself OSS does not give this (but flexibility can become more affordable if OSS provides a bigger community of developers). I think you should remove this from the benefits section - or at least rephrase/recast the idea.

**The revision of that paragraph (above) to highlight the community aspects may now be more convincing. We agree any OSS project has little requirement for flexibility of configuration. However, as mentioned by the referee, as a community develops around a model, users will wish to adapt the model to be able to handle different applications. This then increases the models flexibility of configurations.**

P1 L29 key features: You later describe how CLASSIC provides a means to check that results are reproducable (checksums etc.), which could just about fall under your 8th point here (benchmarking). To me this is an important aspect in its own right and might deserve explicit mention here.

**Thanks for the suggestion. We have amended, "8) extensive benchmarking" to, "8) extensive benchmarking of model state (via checksums) and performance"**

P3 L30 'standard offline model setup' - what is this and where can a user find it (and other configurations)? Later you mention that various runs can be set up and performed using the information provided as part of the package/container, but in general I was unclear as to where (if at all) standard configurations can be obtained. Versioncontrolled code and the container environment are important, but obviously the model set up is also critical. At some point in the manuscript I would like to see some (or clearer) coverage of where standard configurations can be acquired. On P3 it might suffice to point to this information, e.g. "see Section X".

**If the steps in Appendix A are followed, the model source code, all model inputs and configuration files, and the container can be used to produce all results presented here. In these files is the standard model configuration. We agree that model setup is vital for correct model performance (Easterbrook's configurability as mentioned in the Introduction) which is why we have provided all model scripts needed to exactly reproduce our work. Additionally, each user will have 31 examples of how the model was setup for the FLUXNET sites used in our benchmarking suite. To make this more clear we have added the following text to Section 2.1, "the standard model setup is provided for each of the 31 FLUXNET sites; please see Appendix A"**

P9 L7-18: The level of detail and discussion about checksums is unnecessary. The basic principles are well-established and can (most likely) be cited. Most or all of these lines can be removed.

**We agree there is nothing earthshaking here. However we anticipate this paper being read by grad students new to our model, and indeed to modelling in general. As a result, since it is relatively short, we would like to retain it for that future audience**

P9 Sec 3.2.3: I had a quick look at some of the material on Zenodo and it is possible that I didn't find the correct material, but I am wondering if there is any collated scientific documentation for CLASSIC. (In fact I'm thinking of science equations rather than the more code-specific documentation that is the focus of this section. I suspect the answer is 'no', as is the case for many models. Section 2 outlines the science and provides various references that can be followed up, but for many models it takes a keen student and days of work to assemble the definitive description of the science in the latest version of the model. As far as I can see the self-documenting code is very useful but does not address the more detailed scientific documentation. Are there any plans for further documentation? As I said, many models face similar challenges, so this is not a particular criticism of CLASSIC alone!)

**Actually all model documentation can be found in the code and does indeed contain the scientific basis, i.e. equations, for much of the model subroutines. The documentation within the CLASSIC source code can be found in documentation/html/index.html. It is also available from the CLASSIC webpage (e.g. the competition parameterization: https://cccma.gitlab.io/classic/namespacecompetitionscheme.html)**

P10 L25-end: The examples of how to select domains are unnecessary detail in such an overview paper; it is sufficient to know that the capability exists. I suggest removing this.

**Agreed, these have been removed.**

P11 L7-8: I don't understand this. It reads as if FORTRAN is used to convert namelist files into netCDF. Perhaps the ASCII files listed in the namelists are covered into netCDF? If so, rephrase.

**It is correct as written. A Fortran namelist file is used to specify all model initilization values, then a Fortran program reads the namelist and produces a netCDF file that CLASSIC can ingest for model initialization. The conversion script could have been written in another language, but, for simplicity and code reuse, we used Fortran. Most strictly, a namelist file is an ASCII file that follows a formatting convention which Fortran can easily parse. The legacy ASCII files we mentioned were both free and fixed format files that did not correspond to the namelist formatting rules and hence were considerably more difficult to work with.**

P13 L22: I'm only moderately familiar with ILAMB but I thought it mainly (or solely) used colours to show rankings of results (rather than whether thresholds are passed). Also, even more pedantically, I think ILAMB and other tools can be very useful when assessing and presenting how different versions or configurations of a single model compare - in which case your objection to its use for a single model (L23) would not hold.

There we reference Figure 1 from Collier et al., 2018 which uses 'stoplight' colours to denote different thresholds of model performance. We agree that benchmarking tools such as iLAMB and AMBER are very useful for single model comparisons. However, our specific objection in the line mentioned is the use of colour schemes that convey the existence of important thresholds in the score values, as would be implied by the use of 'stoplight' colours.

P6 L13: Being able to use the same code in standalone and coupled applications is, of course, highly advantageous. I know other models also struggle to ensure that a single code base can be used for both (with the main difficulties often being at the higher-level interfaces to parent structures). To me this is an important consideration that could (possibly) be brought up earlier in the manuscript (perhaps as an important aspiration for CLASSIC, even if in fact it is still in the future). As it is, some readers might be slightly confused by the earlier claims that the models are used in coupled applications, yet at this late stage they are told the codes differ (even if in fact the differences are largely 'technical' rather than 'scientific').

**Yes, it is indeed not simple to have complete coherence between the offline and coupled model codes. To be clear we have added the following, "While future releases of CanESM will include CLASSIC, CanESM5 contains older model versions (CLASS v.3.6.2 and CTEM v. 1.1)." to Section 6.**

**Minor comments**

P3 L10 'second generation' - I never like these terms as they are so vague, but it's just about OK as you quote it! I can't recall who first coined the term (Sellers, Shuttleworth?) but perhaps you could consider a citation (or explanation) to back it up.

**True, it can be confusing. Here we add the definition provided by the creator of CLASS, "The Canadian Land Surface Scheme (CLASS) was initiated in 1987 to produce a 'second generation' land surface scheme, characterized by more soil moisture and thermal layers with a separate treatment of the vegetation canopy (Verseghy, 2000), for inclusion in the Canadian general circulation model (GCM) and has been under continual development since."**

P5 L14 "nine PFTs that map directly onto…": Is it that the 9 PFTs of CTEM are collapsed onto the 4 PFTs used by CLASS, so that two or more CTEM PFTs are represented by one CLASS PFT? This could be made clearer. I assume there is some motivating evidence that supports that this is a reasonable approach that saves CPU time.

**Yes, that is the case. The reasoning was that the physics does not need to know if, for e.g. the broadleaf tree has an evergreen or deciduous habit (its LAI/PAI would convey that information indirectly). While there is a small computational cost savings, this feature comes largely from the model development history.**

P5 L28 fire: I realise Fig.1 is just a schematic, but possibly some representation of fire (and land use) could be included.

**The figure has been revised to include fire and land use change.**

P8 I found the order of some sections in this area was slightly unexpected. Looking back, perhaps it was mainly that I thought the containerisation section might come later, after code design etc., as it is essentially a way to bundle everything that has been created/described previously. Coding standards and docuemenation I would also push up the list. The move to Git/Gitlab I would describe before other details such as how to add new PFTs. Similarly, P9 L20 mentions the move away from fixed-form FORTRAN… but mentally I have already checked outputs using checksums in the previous section. To me anything about the code repository and coding standards comes before checksums. But really this is personal preference, and your order is fine!

**Yes, arguably there could be many different ways to organize these sections. Since the referee allows this to fall under personal preference, we would prefer to retain the original structure.**

P10 L20: As I read this, point runs can produce gridded (2-D) outputs. Is that really the case?

**No, it isn't. To correct this we have amended this sentence from, "To run the model at a point location, the model input and output files can be either, also site-level with point-scale information, or regional (two-dimensional fields)." to, "To run the**

**model at a point location, the model input files can be either, also site-level with point-scale information, or regional (two-dimensional fields). "**

P11 L22 and others: I'd prefer "3 or 6" rather than "3/6" which I keep reading as a fraction. I assume CLASSIC is fairly flexible in terms of timestep lengths and data intervals, but this section (and maybe others) does read a bit as if only certain timesteps are allowed. e.g. L21 says inputs are typically 3 or 6 hourly (which is fine), but later in the paragraph we are talking about "3/6" hourly data and a 30 minute timestep - i.e. it becomes specific. Consider clarifying that the model is indeed more flexible than this.

**Yes, the model is indeed more flexible. While we generally use the offline model on a 30 minute physics timestep with 3 or 6 hourly meteorology, any combination where the meteorology is some multiple of the physics timestep could work, such as a 15 minute physics timestep with 2 hour meteorology. We have rephrased this section to make this more apparent, "For global offline simulations, reanalysis meteorological variables are typically available on either a 3 hour or 6 hour timestep, but it could be any multiple of the physics timestep. To convert the reanalysis meteorological data to the offline CLASSIC physics timestep, CLASSIC disaggregates the coarse temporal resolution meteorology on the fly. Surface pressure, specific humidity, wind speed and surface temperature are linearly interpolated. Long-wave radiation is uniformly distributed across the reanalysis time period. Shortwave radiation is distributed diurnally using the day of year and a grid cell's latitude with the maximum value occurring at solar noon. The total reanalysis time period precipitation amount determines the number of wet half-hours in each period following(Arora, 1997). In a conservative manner, the total time period's precipitation amount is then randomly spread across the wet physics timestep periods. If CLASSIC is being run with observed meteorology, such as from an eddy covariance tower, on a 15 - 30 minute timestep, the meteorology is used without modification."**

P12 Sec4.2: Consider clarifying at this stage that AMBER is included in the container (as is later made clear).

**We have changed "All AMBER outputs from site-level evaluation of CLASSIC v. 1.0 are included in the model benchmarking archive (Table 2)." at the end of Section 4.2 to "All AMBER outputs from site-level evaluation of CLASSIC v. 1.0 are in the model benchmarking archive (Table 2) with AMBER itself included in the CLASSIC software container."**

**Typos etc.**

**For all typos listed, we have either adopted the change or defer to the GMD copy editor for whether the change is needed.**

P3 L31: add a comma after 61m.

P10 L20: 'also' seems to be misplaced here. Remove?

P11 L4: remove 'which'

P12 L12: remove 'and'

P13 L21 (and at least one other location): change "don't" to "do not"

P14 L1 'as well': I'd prefer 'additionally'.

P15 L11: change 'Both' to 'both'

P18 L24: Misplaced comma (and I'd remove the first comma on L25 too).

P39 L38: I haven't pored over all the references, but by chance I noticed a slightly dubious entry here: '0, null'.

**Anonymous Referee #2**

Moving CLASS-CTEM to an open source community model is an important initiative. This paper is a well-written discussion – perhaps a little unbalanced in the level of technical detail between different sections and with a few typos to pick up in copy editing, but those are not major problems. I judge that is acceptable for publication essentially as it stands, but I have some minor suggestions and questions:

**Thank you for your review**

- soil freezing and fire are mentioned in the text but not represented in Figure 1.

**We have revised the figure to include them**

- can separate reference heights be provided for temperature and wind speed? They often differ, both in site measurements and global datasets - total precipitation is required as an input, but what flexibility is there in specifying how the model will divide it into snow and rain?

**Yes in both cases. We have now added the following sentence to the Section 2.1 (Model physics: CLASS), "Differing measurement heights for the observed temperature and wind speed are accounted for. Since the forcing data typically does not separate precipitation into its rainfall and snowfall components, three options are available in the CLASSIC modelling framework, 1) a strict threshold of 0 $^\circ$C below which all precipitation is considered snowfall, 2) a linear gradual transition of rainfall into snowfall as temperature decreases from 2 $^\circ$C to 0 $^\circ$C, and 3) a non-linear transition from rainfall to snowfall as temperature decreases from 6 $^\circ$C to 0 $^\circ$C. Here, we use option 1."**

- the single model snow layer is a historical feature of CLASS, but is there a scientific justification for maintaining it in a model with many more ground layers?

**We have retained the single snow layer in our physics scheme as it continues to perform well. While outside the scope of this manuscript, early results from SnowMIP have indicated CLASS (CLASSIC's physics model predecessor) is amongst the top performing models at numerous field sites studied (P. Bartlett pers. comm. 2020).**

- other models have used FLUXNET data for benchmarking. Could a short discussion of CLASSIC performance in the context of other studies be added

**While other models have used FLUXNET to benchmark their model, we don't feel it is straight-forward to compare across the studies. Other studies may have tuned their model performance to the sites run, run only a small handful of sites, or been interested in only one aspect of model performance for a particular study making comparison difficult or impractical. To acknowledge the extensive use of FLUXNET in model evaluation, we have added some example studies that have used FLUXNET data to Section 4.1, "FLUXNET data has been used extensively to improve and evaluate land surface models hydrology, energy, and carbon fluxes as well as model processes (e.g., Blyth et al., 2010, Stockli et al. 2008, Melaas et al., 2013)"**